# SkiLD: Unsupervised Skill Discovery Guided by Factor Interactions

**Zizhao Wang**[1*] **Jiaheng Hu**[1*] **Caleb Chuck**[1*] **Stephen Chen**[1]
**Roberto Martín-Martín**[1] **Amy Zhang**[1] **Scott Niekum**[2] **Peter Stone**[1,3]
[1]University of Texas at Austin    [2]University of Massachusettes, Amherst    [3]Sony AI
{zizhao.wang,jiahengh,stephen.chen,robertomm}@utexas.edu
amy.zhang@austin.utexas.edu, sniekum@umass.edu, {calebc,pstone}@cs.utexas.edu

## Abstract

Unsupervised skill discovery carries the promise that an intelligent agent can learn reusable skills through autonomous, reward-free environment interaction. Existing unsupervised skill discovery methods learn skills by encouraging distinguishable behaviors that cover diverse states. However, in complex environments with many state factors (e.g., household environments with many objects), learning skills that cover all possible states is impossible, and naively encouraging state diversity often leads to simple skills that are not ideal for solving downstream tasks. This work introduces Skill Discovery from Local Dependencies (SkiLD), which leverages state factorization as a natural inductive bias to guide the skill learning process. The key intuition guiding SkiLD is that skills that induce **diverse interactions** between state factors are often more valuable for solving downstream tasks. To this end, SkiLD develops a novel skill learning objective that explicitly encourages the mastering of skills that effectively induce different interactions within an environment. We evaluate SkiLD in several domains with challenging, long-horizon sparse reward tasks including a realistic simulated household robot domain, where SkiLD successfully learns skills with clear semantic meaning and shows superior performance compared to existing unsupervised reinforcement learning methods that only maximize state coverage. Code and visualizations are at `https://wangzizhao.github.io/SkiLD/`.

## 1 Introduction

Reinforcement learning (RL) achieves impressive successes when solving decision-making problems with well-defined reward functions [65, 20, 33]. However, designing this reward function is often not trivial [7, 61]. In contrast, humans and other intelligent creatures can learn, without external reward supervision, behaviors that produce repeatable and predictable changes in the environment [18]. These behaviors, which we call *skills*, can be later repurposed to solve downstream tasks efficiently. One of the promises of this form of unsupervised RL is to endow artificial agents with similar capabilities to discover reusable skills without explicit rewards.

One predominant strategy of prior skill discovery methods focuses on training skills to reach diverse states while being distinguishable [19, 59, 50]. However, in complex environments that contain many *state factors*—distinct elements such as individual objects in a household (a formal description in Sec. 2.1), the exponential number of distinct states makes it impossible to learn skills that cover every state. Consequently, these methods typically result in simple skills that only change the easy-to-control factors (e.g., in a manipulation task moving the agent itself to diverse positions or

---

*Indicates equal contribution

38th Conference on Neural Information Processing Systems (NeurIPS 2024).

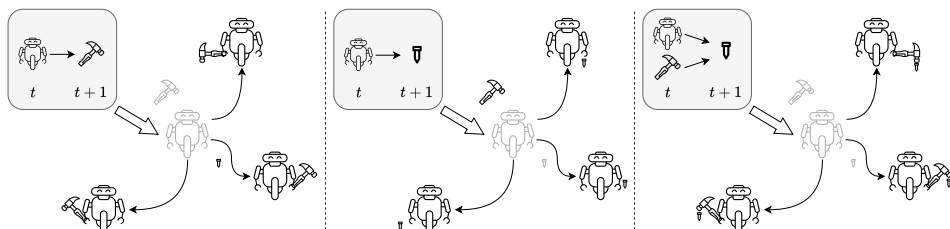

Figure 1: **Skill Discovery from Local Dependencies** (SkiLD) describes skills that encode interactions (i.e., local dependencies) between state factors. In contrast to prior diversity-based methods that can easily get stuck by moving the robot to diverse, but non-interactive states, and factor-based methods that are trained to manipulate the hammer and nail, but not their interactions, SkiLD not only manipulate each object (left, middle) but also induce interactions between them (right), by specifying different local dependencies. These skills are often more useful than the "easy" skill learned by previous methods for downstream task-solving.

manipulating each factor independently), and fail to cover other desirable but challenging behaviors. Meanwhile, in a factored state space, many downstream tasks require inducing interactions between state factors, e.g., cooking requires using a knife to cut the ingredients and cooking them in a pan, etc. Unsurprisingly, these simple skills often struggle to solve such tasks, resulting in poor downstream performance.

Our key insight is to utilize interactions between state factors as a powerful inductive bias for learning useful skills. In factored state spaces and their downstream tasks, there usually exist bottleneck states that an agent must pass through to explore different regions of the environment, and many of them can be characterized by interactions between state factors. For example, in a household environment, a robot must first grasp the knife before moving it to different locations, with the bottleneck being the interaction between the robot and the knife. In environments that have a large state space due to many state factors, rather than inefficiently relying on randomly visiting different states to reach such bottlenecks, we propose to train the agent to actively induce these critical interactions.

To this end, we introduce Skill Discovery from Local Dependencies (SkiLD), a novel skill discovery method that explicitly learns skills that induce diverse interactions. Specifically, SkiLD models the interactions between state factors using the framework of *local dependencies* (where local refers to state-specific, see details in Sec. 2.2) and proposes a novel intrinsic reward that 1) encourages the agent to induce specified interactions, and 2) encourages the agent to discover diverse ways of inducing specified interaction, as visualized in Figure 1. During skill learning, SkiLD gradually discovers new interactions and learns to induce them, based on the skills that it already mastered, resulting in a diverse set of interaction-inducing behaviors that can be readily repurposed for downstream tasks. During task learning, the skill policy is reused, and a task-specific policy is learned to select (a sequence of) skills to maximize task rewards efficiently.

We evaluate the performance of SkiLD on factor-rich environments with 10 downstream tasks against existing unsupervised reinforcement learning methods. Our experiments indicate that SkiLD learns to induce diverse interactions and outperforms other methods on most of the examined tasks.

## 2 Background

In this paper, our unsupervised skill discovery method is set up in a factored Markov decision process and builds off previous diversity-based methods, as described in Sec. 2.1. To enhance the expressivity of skills, our method further augments the skill representation with interactions between state factors, which we formalize as local dependencies as described in Sec. 2.2.

### 2.1 Factored Markov Decision Process (Factored MDP)

We consider unsupervised skill discovery in a reward-free Factored Markov Decision Process [8] defined by the tuple $\mathcal{M} = (\mathcal{S}, \mathcal{A}, p)$. $\mathcal{S} = \mathcal{S}^1 \times \cdots \times \mathcal{S}^N$ is a factored state space with $N$ subspaces, where each subspace $\mathcal{S}^i$ is a multi-dimensional continuous or discrete random variable. Then, correspondingly, each state $s \in \mathcal{S}$ consists of $N$ state factors, i.e., $s = (s^1, \ldots, s^N), s^i \in \mathcal{S}^i$. In this paper, we use uppercase letters to denote random variables and lowercase for their specific values (e.g.,

$S$ denotes the random variable for states $s$). $\mathcal{A}$ is the action space, and $p$ is an unknown Markovian transition model that captures the probability distribution over the next state $S' \sim p(\cdot|S, A)$.

The factorization in $\mathcal{S}$ inherently exists in many environments, and is a common assumption in prior unsupervised skill discovery works [22, 29, 27]. For example, in robotics, an environment typically consists of a robot and several objects to manipulate, and, for each object, $S^i$ would represent its attributes of interest, like pose. In this work, we explore how we can utilize a given state factorization to improve unsupervised skill discovery. In practice, the factorization can either be directly provided by the environment or obtained from image observations with existing disentangled representation learning methods [47, 31].

Following prior work, our method consists of two stages—skill learning and task learning. During the skill learning phase, we seek to learn a skill policy $\pi_\omega(\cdot|s, z)$, which defines a conditional distribution over actions given the current state $s$ and some skill representation $z$, where skills indicate the desired behaviors of the agent. Once the skills are learned, they can be chained together to solve downstream tasks during the task learning phase through an extrinsic reward-optimizing policy. During task learning, a downstream task reward function $r : \mathcal{S} \times \mathcal{A} \to \mathbb{R}$ is provided by the environment. A high-level policy $\pi(z|s)$ is then trained to optimize the expected return through outputting correct skills $z$ given state $s$.

## 2.2 Identifying Local Dependencies between State Factors

A key insight of SkiLD is to utilize interactions between state factors (or, formally, local dependencies) as part of the skill representation. In later sections, these local dependencies are compiled into a binary matrix $\mathcal{G}(s, a, s') = \{0, 1\}^{N \times (N+1)}$ representing the local dependencies between all factors. In this section, we first formally define local dependencies, introduce their identification, and finally discuss their application to factored MDPs.

SkiLD takes a causality-inspired approach for defining and detecting local dependencies [6, 58], where we use *local* to refer to a particular assignment of values for a random variable, as opposed to *global* which applies to all values. Formally, for an *event of interest* $Y = y$ and its potential causes $X = (X^1, \ldots, X^N)$, given the value of $X = x$, local dependencies focus on which $X^i$s are the state-specific cause of the outcome event $Y = y$ (for simplicity of presentation, in this section we overload $N$ as the number of potential causes rather than number of variables and $p$ as the transition function according to a subset of the variables). Formally, we denote the general data generation process of $Y$ as $p : X \to Y$ and the data generation process when $Y$ is *only influenced* by a subset of $X$ as $p^{\bar{X}} : \bar{X} \to Y$, where $\bar{X} \subseteq X$. Then, given the value of all variables, $X^1 = x^1, \cdots, X^N = x^N$ and $Y = y$, we say $Y$ locally depends on $\bar{X}$, if $\bar{X}$ is the *minimal* subset of $X$ such that knowing their values is necessary and sufficient to generate the result of $Y = y$, i.e.,

$$\underset{\bar{X} \subseteq X}{\arg\min} |\bar{X}| \qquad \text{s.t.} \quad p^{\bar{X}}(Y = y | \bar{X} = \bar{x}) = p(Y = y | X = x), \tag{1}$$

where $|\bar{X}|$ is the number of variables in $\bar{X}$. For example, suppose that a robot opens a refrigerator door in a particular transition. The event of interest $Y = y$ is the refrigerator door becoming open, and it locally depends on two factors: the robot and the refrigerator door, while other state factors such as objects inside the refrigerator do not locally influence $Y$.

To identify local dependencies, one can conduct a conditional independence test $y \perp\!\!\!\perp x^i | \{x/x^i\}$ to examine whether a variable $X^i$ is necessary for predicting $Y = y$. In prior works, one form of this test is to examine whether the pointwise conditional mutual information (pCMI) is greater than 0,

$$\text{pCMI}(y; x^i | \{x/x^i\}) = \log \frac{p(y|x)}{p^{\{X/X^i\}}(y|\{x/x^i\})} > 0. \tag{2}$$

If so, then it suggests that knowing $X^i = x$ provides additional information about $Y$ that is not present in $\{X/X^i\}$, and $Y$ locally depends on $X^i$. As the data generation processes are generally unknown, one has to approximate them with learned models. Recent work in RL has utilized various approximations such as attention weights [53], Granger causality [14], and input gradients [63].

In this work, for a transition $(S = s, A = a, S' = s')$, the event of interest is each next state factor being $(S^i)' = (s^i)'$, and we infer whether it locally depends on each state factor $S^j$ and the action $A$ (i.e., whether there is an interaction between state factors $i$ and $j$, where factor $j$ influences $i$). Then

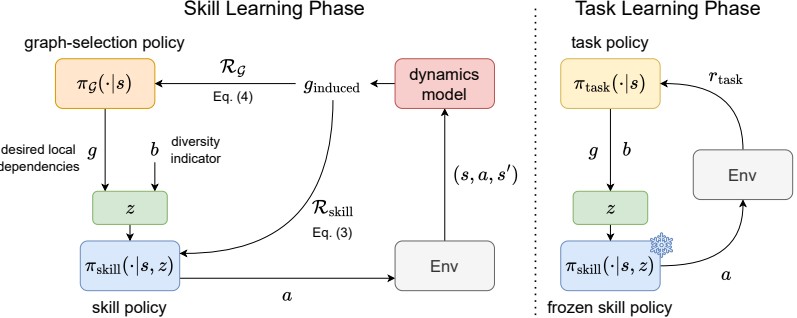

Figure 2: During **skill learning** of SkiLD, the graph-selection policy specifies desired local dependencies for the skill policy to induce, and the induced dependency graph is identified by the dynamics model and used to update both policies. During **task learning** (right), the skill policy is kept frozen and a task policy is trained to select skills to maximize task reward.

we aggregate all local dependencies into a state-specific dependency graph (abbreviated in this work to *dependency graph*). This overall dependency graph is represented with $\mathcal{G}(s, a, s') = \{0, 1\}^{N \times (N+1)}$, and an edge $\mathcal{G}^{ij}(s, a, s')$ denotes, during the transition $(s, a, s')$, that state factor $(s^i)'$ (the "$Y = y$") locally depends on $s^j$ (one of the $X^j$):

$$\mathcal{G}^{ij} := \mathrm{pCMI}((x^i)'; x^j | \{x/x^j\}) > 0 \tag{3}$$

This graph is used to enhance skill representation, as explained in detail in Section 3.

## 3 Skill Discovery from Local Dependencies (SkiLD)

In this section, we describe SkiLD, which enhances skills using local dependencies. SkiLD represents local dependencies as *state-specific dependency graphs*, defined in Sec. 2.2, and learns to induce different dependency graphs in the environment for different skills. To intelligently generate target dependency graphs during training, SkiLD frames unsupervised skill discovery as a hierarchical RL problem described in Fig. 2 and Alg. 1, where a high-level graph selection policy chooses target local dependencies to guide exploration and skill learning, and a graph-conditioned skill policy learns to induce the specified local dependencies using primitive actions.

This framework requires formalizing three components: (1) the skill representation $\mathcal{Z}$, presented in Sec. 3.1, (2) the graph selection policy $\pi_{\mathcal{G}}(z|s)$ and its reward function $\mathcal{R}_{\mathcal{G}}$, presented in Sec. 3.2, and (3) the skill policy $\pi_{\mathrm{skill}}(a|s, z)$ and its corresponding reward function $\mathcal{R}_{\mathrm{skill}}$, presented in Sec. 3.3.

### 3.1 Skill Representation $\mathcal{Z}$

Prior unsupervised skill discovery methods usually focus skill learning on changing the state or each factor diversely, which is inefficient when there exist bottleneck states for explorations. Consequently, they are can be limited to learning simple skills, for example, only changing the easiest-to-control factor in the state (i.e., the agent itself). To address this problem, SkiLD not only focuses on changing the state but also considers the interactions between state factors.

**Skill Representation.** SkiLD represents the skill space as the combination of two components: $\mathcal{Z} = \mathcal{G} \times \mathcal{B}$, where $g \in \mathcal{G}$ is a state-specific dependency graph that specifies the *desired* local dependencies between state factors (e.g., hammering the nail), and $b \in \mathcal{B}$ is a diversity indicator the same as that used in Eysenbach et al. [19]. While the agent inducing particular local dependencies $g$, we use $b$ to further encourage it to visit distinguishable states (e.g., under different $b$ values, training the agent to hammer the nail into different locations). Specifically, the dependency graph is represented as a binary matrix $\mathcal{G} = \{0, 1\}^{N \times (N+1)}$. As described in Sec. 2.2, each edge $\mathcal{G}^{ij}$ denotes, during the transition $(s, a, s')$, whether the state factor $(s^i)'$ locally depends on $s^j$. The diversity indicator $\mathcal{B}$ can be discrete or continuous. In this work, without loss of generality, we follow the procedure of Eysenbach et al. [19] and use a discrete $b$ sampled uniformly from $\{1, \ldots, K\}$, where $K$ is a predefined number.

Given this skill space, SkiLD learns the skills as a skill-conditioned policy $\pi_{\mathrm{skill}} : \mathcal{S} \times \mathcal{Z} \to \mathcal{A}$, where $\pi_{\mathrm{skill}}$ is trained to reach diverse states while ensuring that local dependencies specified by the graph $g$

**Algorithm 1** SkiLD Skill Discovery

---

1: Initialize the high-level graph-selection policy $\pi_{\mathcal{G}} : \mathcal{S} \to \mathcal{G}$, the low-level skill policy $\pi_{\text{skill}}$ : $\mathcal{S} \times \mathcal{Z} \to \mathcal{A}$, the diversity indicator discriminator $q : \mathcal{S} \times \mathcal{G} \to p(\mathcal{B})$, and the dynamics model $f : \mathcal{S} \times \mathcal{A} \to \mathcal{S}$, graph selection interval $L$.
2: **for** each skill training timestep $i$ **do**
3:     *// data collection*
4:     **if** $i \% L == 0$ **then**
5:         Sample the target dependency graph $g \sim \pi_{\mathcal{G}}(s)$.
6:         Sample the diversity indicator from uniform distribution $b \sim \text{Uniform}(\mathcal{B})$.
7:         Compose the skill variable $z = (g, b)$.
8:     **end if**
9:     Collect state transitions $(s, z, a, s')$ with actions from $\pi_{\text{skill}}(a|s, z)$.
10:     Infer the induced dependency graph $g_{\text{induced}}(s, a, s')$ using the dynamics model $f$ (Sec. 3.1).
11:     Update the history of the seen graphs with $g_{\text{induced}}(s, a, s')$.
12:     *// training*
13:     Sample a batch of $(s, z, a, s')$ from the replay buffer.
14:     Update the dynamics model $f(s, a)$ by minimize the prediction error w.r.t. $s'$.
15:     Update the high-level policy with reward $\mathcal{R}_{\mathcal{G}}$ (Eq. 5) with the history of seen graphs.
16:     Update the discriminator $q(b|s, g)$ with the discrimination (cross-entropy) loss.
17:     Infer the induced dependency graph $g_{\text{induced}}(s, a, s')$ using the dynamics model $f$ (Sec. 3.1).
18:     Infer the diversity reward $\mathcal{R}_{\text{diversity}} = \log q(b|s, g)$.
19:     Update $\pi_{\text{skill}}$ with $\mathcal{R}_{\text{skill}} = \mathbb{1}[g_{\text{induced}}(s, a, s') = g] \cdot (1 + \lambda \mathcal{R}_{\text{diversity}})$ (Eq. 4).
20: **end for**

---

are induced. Before we describe $\pi_{\text{skill}}$ training in Sec. 3.3, we first discuss how to select the skill $z$ for $\pi_{\text{skill}}$ to follow during the skill learning stage.

## 3.2 High-Level Graph-Selection Policy $\pi_{\mathcal{G}}$

To acquire skills that are useful for downstream tasks, the skill policy $\pi_{\text{skill}}$ needs to learn to induce a wide range of local dependencies *sample-efficiently*. To this end, we propose to learn a graph-selection policy $\pi_{\mathcal{G}} : \mathcal{S} \to \mathcal{G}$ to guide the training of $\pi_{\text{skill}}$. Specifically, training $\pi_{\text{skill}}$ requires a wise selection of graphs — as graph space $\mathcal{G}$ increases super-exponentially in the number of state factors $N$, many graphs are not inducible. To this end, we only select target graphs for the skill policy from a history of all seen graphs. As the agent learns to induce existing graphs in diverse ways, new graphs may be encountered, gradually expanding the set of seen graphs.

However, though this history guarantees graph inducibility, two challenges still remain: (1) How to efficiently explore novel local dependencies, especially hard-to-visit ones? (2) For all seen graphs, which one should $\pi_{\text{skill}}$ learn next to maximize training efficiency? We address these challenges based on the following insight — compared to well-learned skills, $\pi_{\text{skill}}$ should focus its training on underdeveloped skills. Meanwhile, learning new skills opens up the possibility of visiting novel local dependencies, e.g., learning to grasp the hammer makes it possible for the robot to hammer the nail.

According to this idea, we learn a graph-selection policy $\pi_{\mathcal{G}}$ that guides the exploration and training of the skill policy $\pi_{\text{skill}}$. Specifically, $\pi_{\mathcal{G}} : \mathcal{S} \to \mathcal{G}$ selects a new dependency graph the skill policy should induce for the next $L$ time steps. To increase the likelihood of visiting hard graphs, $\pi_{\mathcal{G}}$ is trained to maximize the following graph novelty reward

$$\mathcal{R}_{\mathcal{G}} = \frac{1}{\sqrt{C(g_{\text{visited}})}}, \tag{4}$$

where $C(g_{\text{visited}})$ is the number of times that we have seen the graph in the collected transition.

## 3.3 Low-Level Skill Policy $\pi_{\text{skill}}$

Given the skill parameter $z$ from the graph-selection policy, SkiLD learns skills as a skill-conditioned policy $\pi_{\text{skill}} : \mathcal{S} \times \mathcal{Z} \to \mathcal{A}$, where $\pi_{\text{skill}}$ learns to reach diverse states while ensuring that the local dependencies specified by $g$ are induced. During skill learning, we select actions by iteratively calling

the skill policy $\pi_{\text{skill}}$, and we denote $g_{\text{induced}}(s, a, s')$ as the graph that describes the local dependencies induced in a transition $(s, a, s')$ when executing a selected action $a$. We design the reward function of the skill policy as:

$$\mathcal{R}_{\text{skill}} = \mathbb{1}[g_{\text{induced}}(s, a, s') = g] \cdot (1 + \lambda \mathcal{R}_{\text{diversity}}), \tag{5}$$

where $\mathbb{1}[g_{\text{induced}}(s, a, s') = g]$ measures whether the induced dependency graph matches the desired graph, $\mathcal{R}_{\text{diversity}}$ is the weighted diversity reward that further encourages visiting diverse states when the desired graph is induced, and $\lambda$ is the coefficient of diversity reward. In the following paragraphs, we describe how we infer $g_{\text{induced}}(s, a, s')$ and estimate $\mathcal{R}_{\text{diversity}}$ for each transition.

**Inferring Induced Graphs.** To infer the induced graph for a transition $(S = s, A = a, S' = s')$, we need to determine, for each $(\mathcal{S}')^i$, whether it locally depends on each factor $\mathcal{S}^j$ and the action $\mathcal{A}$. Following Sec. 2.2, we evaluate the conditional dependency $(s^i)' \not\perp s^j | \{s/s^j, a\}$ by examining whether their pointwise conditional mutual information (pCMI) is greater than a predefined threshold $\epsilon$. If $\text{pCMI}^{ij} = \frac{p((s^i)'|s,a)}{p((s^i)'|\{s/s^j,a\})} \geq \epsilon$, it suggests that $s^j$ is necessary to predict $(s^i)'$ and thus the local dependency exists. Meanwhile, as the transition probability $p$ is unknown, we approximate it with a learned dynamics model that is trained to minimize prediction error.

Finally, after obtaining the induced dependency graph, we evaluate $\mathbb{1}[g_{\text{induced}}(s, a, s') = g]$ by examining whether each edge $g_{\text{induced}}^{ij}$ matches the corresponding edge in the desired graph $g^{ij}$. As $\mathcal{R}_{\text{skill}}$ only provides sparse rewards to the skill policy when the desired graph is induced, we use hindsight experience replay [2] to enrich learning signals, by relabelling induced graphs as desired graphs in some episodes.

**Diversity Rewards.** When the skill policy induces the desired graph, $\mathcal{R}_{\text{diversity}}$ further encourages it to visit different distinguishable states under different diversity indicators $b$, e.g., hammering the nail to different locations. This diversity enhances the applicability of learned skills. To this end, we design the diversity reward $\mathcal{R}_{\text{diversity}}$ as the forward mutual information between visited states and the diversity indicator $I(s; b)$, following DIAYN. To estimate the mutual information, we approximate it with a variational lower bound $I(s; b) \geq \mathbb{E}_{b,s} \log q(b|s)$, where $q(b|s)$ is a neural network discriminator trained to predict the diversity indicator $b$ from the visited state.

In practice, rather than learning a single low-level skill to handle all graphs, SkiLD utilizes a factorized lower-level policy. When the target dependency graph is specified, SkiLD identifies which state factor should be influenced and uses its corresponding policy to sample primitive actions. More details about this subdivision can be found in Appendix A.

### 3.4 Downstream Task Learning

In SkiLD, after the skill learning stage, we utilize hierarchical RL to solve reward-supervised downstream tasks with the discovered skills. The skill policy, $\pi_{\text{skill}}$ acts as the low-level policy while a task policy, $\pi_{\text{task}} : \mathcal{S} \rightarrow \mathcal{Z}$, is learned to select which skill $z = (g, b)$ to execute for $L$ steps. Compared to diversity-based skills that are limited to simple behaviors, our local-dependency-based skills enable a wide range of interactions between state factors, leading to more efficient exploration and superior performance of downstream task learning.

## 4 Experiments

In this section we aim to provide empirical evidence towards the following questions: **Q1)** Do the skills learned by SkiLD induce a diverse set of interactions among state factors? **Q2)** Do the skills learned by SkiLD enable more efficient downstream task learning compared to other unsupervised reinforcement learning methods? Our learned skills are visualized at `https://sites.google.com/view/skild/`.

### 4.1 Domains

In this work, we focus on addressing the challenge of vast state space brought by the number of state factors. Hence, we evaluate our method on two challenging *object-rich* embodied AI benchmarks: Mini-behavior [32] and Interactive Gibson [42].

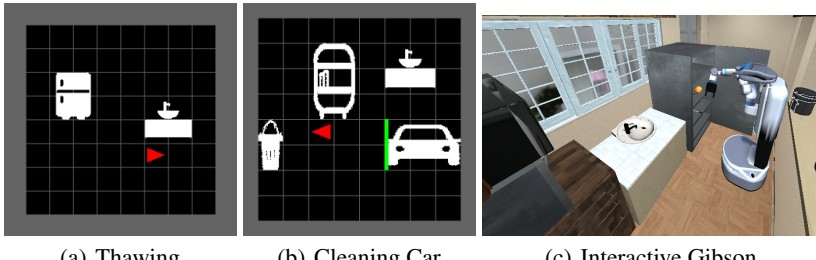

|(a) Thawing|(b) Cleaning Car|(c) Interactive Gibson|

Figure 3: **Evaluation domains**: Mini-behavior: Installing Printer, Thawing and Cleaning Car, and iGibson.

The **Mini-behavior (Mini-BH) domain** [32] (Figure 3a) contains a set of gridworld environments where an agent can move around and interact with a variety of objects to accomplish certain household tasks. While conceptually simple, due to highly **sequentially interdependent** state factors (see details in the Appendix), this domain has been shown to be extremely challenging for the agent's exploration ability, especially under sparse reward [32]. Each Mini-BH environment contains different objects and different success criteria. We tested on three particular environments in Mini-behavior, including:

- **Installing Printer**: A relatively simple environment with three state factors: the agent, a table, and a printer that can be installed.
- **Cleaning Car**: An environment where the objects have rich and complex interactions. The state factors include the agent, a toggleable sink, a piece of rag that can be soaked in the sink, a car that the rag can clean, a soap and a bucket which can together be used to clean the rag.
- **Thawing**: An environment with lots of movable objects. The state factors include the agent, a sink, a fridge that can be opened, and three objects that can be thawed in the sink: fish, olive, and a date.

The **Interactive Gibson (iGibson)** domain [43] (Figure 3b) contains a realistic simulated Fetch Robot that operates in a kitchen environment with a refrigerator, sink, knife, and peach. The peach can be washed or cut. This domain is very difficult especially when using low-level motor commands because much of the domain is free space, meaning that only a minute fraction of action sequences will manipulate the objects meaningfully.

Both Mini-BH and iGibson require learning long-horizon policies spanning many low-level actions from sparse reward, making these challenging domains (see details in Appendix).

## 4.2 Baselines

Before evaluating the empirical questions, we provide a brief description of the baselines. These baselines include unsupervised skill learning, and causal and hierarchical methods.

**Diversity is all you need** (DIAYN [19]): This method learns unsupervised state-covering skills using a mutual information objective. SkiLD utilizes a version of this for state-diversity skills modulated by a desired dependency graph. This baseline determines how incorporating graph information affects the algorithm.

**Controllability-Aware Skill Discovery** (CSD [50]): Extends DIAYN with a factorization based on controllability. This baseline is a comparable skill learning method that leverages state factorization but does not encode local dependencies.

**Exploration via Local Dependencies** (ELDEN [63]): This method utilizes gradient-based techniques to infer local dependencies for exploration. However, without a skill learning component, it can struggle to chain together complex behavior.

**Chain of Interaction Skills** (COInS [14]): This is a hierarchical algorithm that constructs a chain of skills using Granger-causality to identify local dependencies. Because it is restricted to pairwise interactions, it struggles to represent the rich policies necessary for these tasks.

**Vanilla RL**: This baseline uses PPO [57] to directly train an agent with the extrinsic reward. Unlike other baselines, this method does not have a pertaining phase. Since all the task rewards are sparse and the tasks are often long horizon, vanilla RL often struggles.

## 4.3 Interaction Graph Diversity

We first evaluate whether SkiLD is indeed capable of achieving complex interaction graphs (Q1), comparing against two strong skill discovery baselines introduced earlier: DIAYN and CSD.

Each of these methods is trained for 10 Million steps without having access to any reward. Then to evaluate their learned skills, we unroll each of them for 500 episodes with randomly sampled skills $z$ and examine the diversity of the interaction graphs they can induce. Figure 4 illustrates the percentages of episodes where some hard local dependencies have been induced at least once, in Mini-BH Cleaning Car (for simplicity of presentation, see Appendix for results on all inducible local dependency graphs and their meanings). We find that DIAYN and CSD are limited to skills that only manipulate one object individually, for example, picking up the rag (agent, rag, action → rag) or the soap (agent, soap, action → soap). By contrast, SkiLD learns to induce more complicated causal interactions, such as soaking the rag in the sink (sink, rag → rag) and cleaning the car with the soaked mug (car, rag → car).

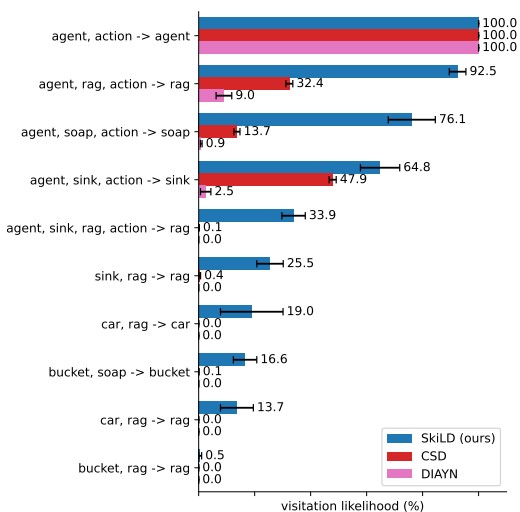

Figure 4: The percentage of episodes where a dependency graph is induced through random skill sampling. Standard deviation is calculated across five random seeds.

## 4.4 Sample Efficiency and Performance

Next, we evaluate whether the local dependency coverage provided by SkiLD leads to a performance boost in downstream task learning under the same number of environment interactions (Q2). We follow the evaluation setup in the unsupervised reinforcement learning benchmark [38], where for a given environment, an agent is first pre-trained without access to task reward for $K_{\text{pt}}$ steps, and then finetuned for $K_{\text{ft}}$ steps. Importantly, the same pre-trained skills are reused on multiple distinct downstream tasks within the same environment, so that only the upper-level skill-selection policy is task-specific. We have $K_{\text{pt}} = 2M$, $K_{\text{ft}} = 1M$ for installing printer, $K_{\text{pt}} = 10M$, $K_{\text{ft}} = 5M$ for thawing and cleaning car, and $K_{\text{pt}} = 4M$, $K_{\text{ft}} = 2M$ for iGibson, and evaluate each method for each task across 5 random seeds. Hyperparameter details can be found in Appendix D. Specifically, we evaluate on the following downstream tasks:

- **Installing Printer**: We have a single downstream task in this environment, where the agent needs to pick up the printer, put it on the table, and turn it on.

- **Thawing**: We have three downstream tasks: thawing the fish or the olive or the date.

- **Cleaning Car**: We consider three downstream tasks, where each task is a pre-requisite of the following one. The tasks are: soak the rag in the sink; clean the car with the rag; and clean the dirty rag using the soap in the bucket.

- **IGibson**: The tasks for this domain are: grasping the peach, washing the peach in the sink, and cutting the peach with a knife.

After skill learning, we train a new upper-level policy that uses $z$ as actions and is trained with extrinsic reward, as described in Section 3.4. Figure 5 illustrates the improvement of SkiLD as compared to other methods. Without combining dependency graphs with skill learning, other methods struggle with any but the simpler tasks. COInS performs poorly because of its chain structure, which restricts the agent controlling policy from picking up objects. ELDEN's exploration reaches graphs, but without skills struggles to utilize that information in downstream tasks. DIAYN learns skills, but few manipulate the objects, so a downstream model struggles to utilize those skills to achieve meaningful rewards. By comparison, SkiLD achieves superior performance on 9 of the 10 downstream tasks evaluated. In the two hardest tasks which require a very long sequence of precise controls, Clean Rag

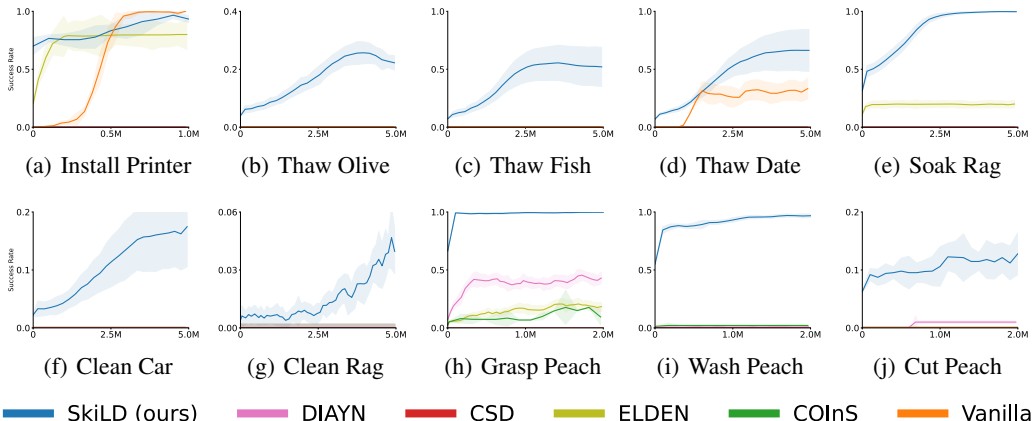

Figure 5: Training curves of SkiLD and baselines on multiple downstream tasks (reward supervised second phase). Each curve depicts the mean and standard deviation of the success rate over 5 random seeds. SkiLD outperforms all baselines for most tasks, converging faster and to higher returns.

and Cut Peach, SkiLD is the only method that can achieve a non-zero success rate (although still far from fully mastering the tasks), showcasing the potential of local dependencies for skill learning.

### 4.5 Graph and Diversity Ablations

We also explore the functionality of the graph and diversity components of the skill parameter $z$ by assessing the downstream performance of SkiLD without these components. This produces two ablative versions of SkiLD: SkiLD without diversity and SkiLD without dependency graphs. To isolate learning from the effect of learned local dependencies, we use ground truth dependency graphs for ablative evaluations where relevant. In Figure 6, learning without graphs results in zero performance, consistent with DIAYN results. In addition, removing diversity produces a notable decline in performance, especially on more challenging tasks like clearning the rag. These evaluations demonstrate that SkiLD benefits from both the incorporation of dependency graphs and diversity.

## 5 Related Work

This work lies in the unsupervised skill learning framework [37], where the agent must discover a set of useful skills which are reward independent. It then extends these skills to construct a 2-layer hierarchical structure [60], where the upper policy receives reward both for achieving novel skills, and can then be tuned to utilize the learned skills to accomplish an end task. Finally, the skills are identified using token causality, a specific problem identified in causal literature.

### 5.1 Unsupervised Skill Learning

This work describes a framework for utilizing local dependency graphs and diversity to discover unsupervised skills. Diversity-based state coverage skills have been explored in literature [19] utilizing forward and backward mutual information techniques to learn a goal space $\mathcal{Z}$, and a skill encoder $q(z|\cdot)$ [11]. This unsupervised paradigm has been extended with Lipschitz constraints [49], contrastive objectives [39], information bottleneck [35], population based methods such as particle estimation [45], quality diversity [44] and mixture of experts [12]. These skills can then be used for hierarchical policies or planners [56, 67, 23], which mirrors the same structure as SkiLD. Unlike these methods, SkiLD adds additional subdivision through dependency graphs, which mitigates the combinatorial explosion of skills that can result from trying to cover a large factored space.

### 5.2 Hierarchical Reinforcement Learning

The hierarchical policy structure in SkiLD where a higher level policy passes a parameter to be interpreted by low-level planners has been formalized in [60], and learned using deep networks utilizing

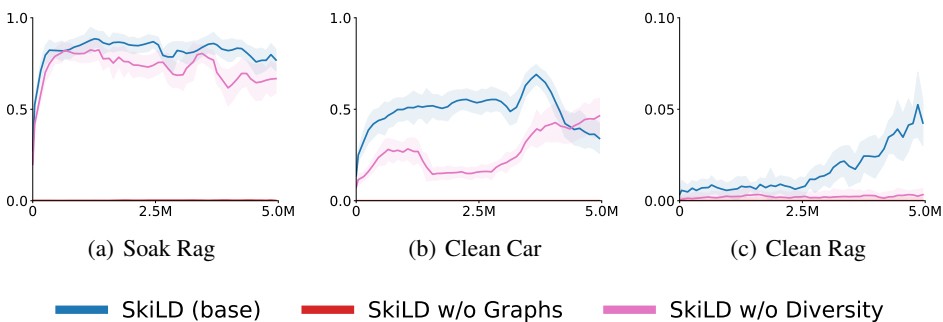

(a) Soak Rag  (b) Clean Car  (c) Clean Rag

——— SkiLD (base)  ——— SkiLD w/o Graphs  ——— SkiLD w/o Diversity

Figure 6: A figure illustrating the ablative performance of SkiLD without diversity or without graphs. Each curve depicts the mean and standard deviation of the success rate over 5 random seeds. Without graphs, the method collapses completely, while removing diversity results in a noticeable reduction in downstream performance.

extrinsic reward [3, 62], attention mechanisms [16], initiation critera [34, 4] and deliberation cost [26]. Hierarchies of goal-based policies [40] has been extended with object-centric representations [66], offline data [48], empowerment [41] and goal counts [51]. In practice, SkiLD uses graph and diversity parameters similar to goal-based methods. However, the space of goals can often be intractable large, and methods to address this use graph laplacians [36] causal chains [13, 14] or general causal relationships [29]. SkiLD is similar to these causal methods but utilizes local dependence along with general two-layer architectures, thus showing increased generalizability.

## 5.3 Causality in Reinforcement Learning

This work investigates the application of local dependency to hierarchical reinforcement learning. This kind of reasoning has been described as "local causality" or "interactions" in prior RL work for data augmentation [53, 54], learning skill chains [13, 14] and exploration [63]. This work is the first synthesis of unsupervised skill learning and local dependencies applied to general 2-layer hierarchical reinforcement learning. Other general causality work investigates action-influence detection [58, 28], affordance learning [10], model learning [30, 21], critical state identification [46], and disentanglement [17]. In the context of relating local dependency and causal inference, we provide a discussion in Appendix C. SkiLD incorporates causality-inspired local dependence to skill learning, resulting in a set of diverse skills.

## 6 Conclusion

Unsupervised skill discovery is a powerful tool for learning useful skills in long-horizon sparse reward tasks. However, many unsupervised skill-learning methods do not take advantage of factored environments, resulting in poor performance in complex environments with several objects. Skill Discovery from Local Dependencies utilizes state-specific dependency graphs, identified using learned pointwise conditional mutual information models, to guide skill discovery. The framework of defining skills according to a dependency graph and diversity goal, combined with a learned sampling scheme, achieves difficult downstream tasks. In domains where hand-coded primitive skills are typically given to the agent, like Mini-behavior and Interactive Gibson, SkiLD can achieve high performance without requiring explicit domain knowledge. These impressive results arise intuitively from incorporating local dependencies as skill targets, illuminating a meaningful direction for unsupervised skill learning to be applied to a wider array of environments.

**Limitations and Future Work** An important assumption of SkiLD is its access to factored state space. While factored state space can often be naturally obtained from existing RL benchmarks and many real-world environments, developments in disentangled representation learning [47, 31] will help with extending SkiLD to unfactored image domains. Secondly, SkiLD requires accurate detection of local dependencies. While off-the-shelf methods [63, 58] work well for detecting local dependencies in our experiments, future works that can more accurately detect local dependencies will be beneficial to the performance of SkiLD.

# 7 Acknowledgement

This work has taken place in the Learning Agents Research Group (LARG) at the Artificial Intelligence Laboratory, The University of Texas at Austin. LARG research is supported in part by the National Science Foundation (FAIN-2019844, NRT-2125858), the Office of Naval Research (N00014-18-2243), Army Research Office (W911NF-23-2-0004, W911NF-17-2-0181), Lockheed Martin, and Good Systems, a research grand challenge at the University of Texas at Austin. The views and conclusions contained in this document are those of the authors alone. Peter Stone serves as the Executive Director of Sony AI America and receives financial compensation for this work. The terms of this arrangement have been reviewed and approved by the University of Texas at Austin in accordance with its policy on objectivity in research.

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

# A    Factored Skills

Learning to reach both a desired graph $g$ and a diversity parameter $b$ through primitive actions is challenging. First, different graphs often have substantially different characteristics, with some graphs that are easy to achieve (eg. action→agent), and others that are quite challenging and rare (eg. agent, knife, fruit→fruit). Not only would it be challenging for a single policy to encode all of these behaviors, the diversity parameter notwithstanding, but over-training the frequency at which certain graphs are called might vary significantly. Rather than trying to learn a single monolithic policy, then, we instead structure the skill parameterized policy $\pi_{\text{skill}}$ as a collection of factored skills: $\pi_{\text{skill},i}$, for each factor $i \in \{1, \dots, N\}$.

This modification to the policy structure results in three changes: **1)** The upper-level action space passes a single row of the graph $\mathcal{G}$, denoted with $g_i$, and the desired factor $i$. **2)** Instead of achieving an entire graph use the achieved row $\mathbb{1}[g_{\text{achieved},i} = g_i]$. **3)** The history of seen graphs $\mathcal{H}$ is replaced with a history of factored graph rows $\mathcal{H}_f$.

Define the history of graph rows as $\mathcal{H}_f := \{\text{unique}\,(i, g_{\text{achieved},i} \quad \forall i \in 1, \dots, N \quad \forall g_{\text{achieved}} \in \mathcal{D})\}$. This takes the unique graph rows from all those seen in previous data. Then the upper policy uses the same historical sampling procedure as with unfactorized graphs: the policy samples discretely from the new history, which will by default return $i, g_i$, a graph row, and the desired factor. This resolves points **1,3**. Point **2** is addressed by replacing Equation 5 with $\mathbb{1}[g_{\text{achieved},i} = g_i]$.

Empirically, we found that without this change, the lower policy rarely learns anything, even simple control of the agent.

# B    Environment Details

In this section, we provide a detailed description of the environment, including its semantic stages representing internal progress toward task completion, state space, and action space. We also highlight that while each task consists of multiple semantic stages, agents do not have access to this information.

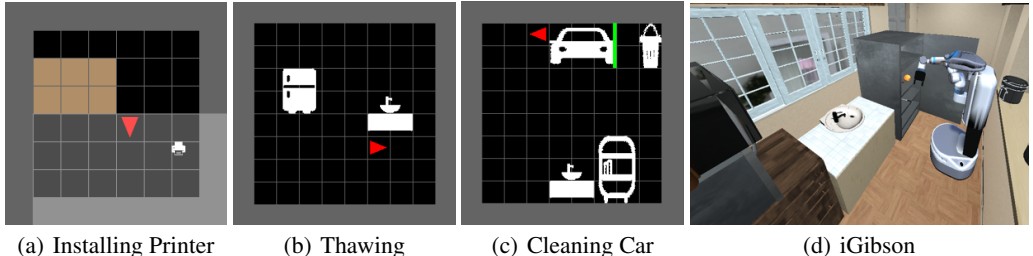

(a) Installing Printer      (b) Thawing      (c) Cleaning Car      (d) iGibson

Figure 7: Environments.

**Installing Printer**    As shown in Fig. 7(a), the Installing Printer environment is relatively simple, consisting of 3 factors: the agent, a printer, and a table. The task requires the agent to complete the following **stages**: (1) pick up the printer, (2) bring the printer to and place it on the table, and (3) turn on the printer. The discrete state space consists of (i) the agent's position and direction, (ii) the positions of the printer and whether it is on or off, and (iii) the position of the table. The discrete action space consists of (i) moving forward, turning left or right, (ii) picking up / placing down the printer, and (iii) turning on / off the printer.

**Thawing**    As shown in Fig. 7(b) and Fig. 10(a), the Thawing environment consists of 6 factors: the agent, a sink, a refrigerator, and three frozen objects: fish, olive, and date. Thawing each object requires the agent to complete the following **stages**: (1) move to and open the refrigerator, (2) take the frozen fish out of the refrigerator, (3) put the fish into the sink, and (4) turn on the sink to thaw it. The discrete state space consists of (i) the agent's position and direction, (ii) the positions of all environment entities, (iii) whether the sink door is turned on, (iv) whether the refrigerator door is opened, and (v) the thawing status of three objects. The discrete action space consists of (i) moving forward, turning left or right, (ii) opening / closing the refrigerator, (iii) turning on / off the sink, and (iv) picking up / placing down each object.

**Cleaning Car**  As shown in Fig. 7(c), the Cleaning Car environment consists of 7 factors: the agent, a car, a sink, a bucket, a shelf, a rag, and a piece of soap. Cleaning both the car and the rag requires the agent to complete the following **stages**: (1) take the rag off the shelf, (2) put it in the sink, (3) toggle the sink to soak the rag up, (4) clean the car with the soaked rag, (5) take the soap off the self, and (6) clean the rag with the soap inside the bucket. The discrete state space consists of (i) the agent's position and direction, (ii) the positions of all environment entities, (iii) whether the sink is turned on, (iv) the soak status of the rag, (v) the cleanness of the rag, and (vi) the cleanness of the car. The discrete action space consists of (i) moving forward, turning left or right, (ii) turning on / off the sink, and (iii) picking up / placing down the rag / soap.

**iGibson**  As shown in Fig. 7(d), the iGibson environment consists of 4 factors: the robot, a knife, a peach, and a sink. The robot can do the following things: (1) grasp peach: move close to the peach and grasp it, (3) wash peach: grasp the peach and place it into the sink, (3) grasp knife: move close to the knife and grasp it, (4) cut peach: grasp the knife and use it to cut the peach. The continuous state space consists of (i) the robot's proprioception, (ii) the poses of all environment entities, and (iii) whether the peach is cut. The continuous action space consists of (i) end-effector position change, (ii) base linear and angular velocity, and (iii) gripper torque (to open/close the gripper).

Though looking conceptually simple, we emphasize that these environments are challenging because of the following factors:

- The state factors are **highly sequentially interdependent**, making skill learning and task learning challenging: for example, in cleaning car environments, the agent can't clean the car until it picks up the rag, turns on the sink, and soaks the rag. These interdependencies between state factors pose great challenges to the agent's exploration ability.

- During the skill-learning stage, we would like the agents to learn **all possible skills** (e.g., manipulating all objects) rather than learning to manipulate a single object.

- During task learning, we use **sparse rewards**, and thus it is further challenging for agents to explore.

- In addition, we use primitive actions, and m**any actions have no effect if it's not applicable in the current state**. So the task is especially challenging for exploration.

## C  Local Dependencies and Causal Inference

In this work, we define local dependencies according to the state factors $X = (X^1, \ldots, X^N)$ and event of interest $Y$, which in the context of an MDP is a subset of the next state factors $X' = (X'^1, \ldots, X'^N)$. In the factored MDP formulation [8], we assume that $p$, the transition dynamics, are represented by a dynamic Bayesian network (DBN) which is a time-directed bipartite graph, with edges only from factors in $X$ to factors in $X'$. In this work, we assume that the underlying ground truth DBN, that is the transition function $p$, can be decomposed according to subsets of state factors $\bar{X}$, such there exists a $p^{\bar{X}}(Y = y | \bar{X} = x)$ for every state.

The factored transition dynamics analogizes with causal inference in the following way: If the state factors and next state factors are each assigned a causal variable by adding the assumption that they can be independently intervened on, and each next state variable carries an associated unobserved noise variable $U^i$, which we assume is independent of any $X^k$ not connected to $X'^j$ and any other next state variable $X'^j$, then we can represent the transition dynamics $p$ with a structural causal model (SCM) [52], a graph connecting the causal variables in $X$ to the causal variables in $X'$.

For a particular outcome variable $Y$ that is one of the next state causal variables $X'$, we can describe local dependence in the RL context according to assumptions about the structural causal model. Represent the non-noise parents of $Y$ as $\mathrm{pa}(Y)$, and the noise parents as $\mathrm{pa}_U(Y)$. Under normal causal assumptions, the structural causal model for $Y$ is a function $f_Y(\mathrm{pa}(Y), \mathrm{pa}_U(Y)) = Y$. Define $\bar{X}$ as a subset of the endogenous parents of $Y$ and $\bar{U}$ as an equivalent subset of the noise variables. Further define the values that $\mathrm{pa}(Y), \mathrm{pa}_U(Y), \bar{X}, \bar{U}$ can take on as $\mathrm{pa}(y), \mathrm{pa}_U(y), \bar{x}, \bar{u}$ respectively, and $(pa(\mathcal{Y})), \bar{\mathcal{X}}, \bar{\mathcal{U}}$ as the set of states the parents of $Y$, the variables in $\bar{X}$ and variables in $\bar{U}$ can take on respectively.

To formalize local invariance, we add the assumption that $f_Y$ can be decomposed into a series of functions $(f_{Y1}(\bar{X}_1 = \bar{x}_1, \bar{U}_1 = \bar{u}_1), \ldots, f_{Yk}(\bar{X}_k = \bar{x}, \bar{U}_k = \bar{u}_k))$ and $g_Y(\mathrm{pa}(Y) = \mathrm{pa}(y), \mathrm{pa}_U(Y) =$

$\text{pa}_U(y))$, where each $f_{Yi} : \bar{\mathcal{X}} \times \bar{\mathcal{U}} \to \mathcal{Y}$ and $g : \text{pa}(\mathcal{Y}) \to \{1, \ldots, k\}$, a function mapping the parents of $Y$ to one of the functions. Then if $f$ is represented as:

$$f(\text{pa}(x), \text{pa}_U(y)) := \sum_{i=1}^{k} \mathbb{1}(g_Y(\text{pa}(y), \text{pa}_U(y)) = i) f_{Yi}(\bar{x}_i, \bar{u}_i) \tag{6}$$

The local dependence of $Y = y$ in a particular state $(x, x')$ is then the set of variables in $\bar{X}_i$ for the particular $i$ where $\mathbb{1}(g_Y(\text{pa}(y), \text{pa}_U(y)) = i) = 1$, and the pCMI test is a way of uncovering these local dependencies from observational data.

Local dependence has been investigated in the field of context-specific independence [55, 9], which seeks to find particular assignments of a subset of the causal variables under which an outcome is independent of some subset of the inputs. In particular, context-set specific independence [9] determines if a variable is independent of other variables on a particular subset of states, described as the partial context set. While our work uses the pCMI test described in Equation 2, context-specific independence focuses on complete independence using knowledge of the structural model.

Alternatively, interactions can be viewed as the causes ($\bar{X}$) of particular effects ($Y$), which have also been investigated under the description of token or actual cause [25] (as opposed to general cause). Actual cause utilizes a series of counterfactual tests to determine if a cause is necessary, sufficient, and minimal for an outcome. Actual cause has primarily been applied in simple, discrete examples [5, 24], making it difficult to directly apply to RL. However, recent work has incorporated the notion of context-specific independence and extended actual cause to more complex domains [15].

## D   Implementation Details

The hyperparameters of skill learning and task learning can be found in Table 1. As it is challenging to identify local dependencies using learned dynamics models in Thawing and iGibson environments, we use ground truth local dependencies from simulator. The codebase is built on tianshou [64] for backend RL, though with significant modifications.

The 5 seeds selected are 0 - 4. The experiments were conducted on machines of the following configurations:

- Nvidia A40 GPU; Intel(R) Xeon(R) Gold 6342 CPU @2.80GHz
- Nvidia A100 GPU; Intel(R) Xeon(R) Gold 6342 CPU @2.80GHz

## E   Additional Results

### E.1   Interaction Graph Diversity

Figure 8 illustrates the percentages of episodes where all local dependencies have been induced at least once, in Mini-BH Cleaning Car. Again, SkiLD (ours) induces all inducible dependency graphs, while baselines fail to induce hard graphs with challenging pre-conditions.

The meaning of the graphs are:

- agent, action $\to$ agent: agent moving.
- agent, rag, action $\to$ rag: agent picking up the rag or moving it.
- agent, soap, action $\to$ soap: agent picking up the soap or moving it.
- agent, X $\to$ agent: X blocking the agent's motion.
- agent, sink, action $\to$ sink: agent turning on or off the sink.
- agent, sink, rag, action $\to$ rag: agent soaking the rag in the sink (which requires that the sink is turned on).
- sink, rag $\to$ rag: the same as above.
- car, rag $\to$ car: the rag cleaning the car and getting dirty (which requires that the rag is soaked).
- car, rag $\to$ rag: the same as above.

Table 1: Parameters of Skill Learning and Task Learning. Parameters shared if not specified.

| Name | | Environments | | |
|---|---|---|---|---|
| | | Printer | Thawing Cleaning Car | iGibson |
| Skill Policy | algorithm | | Rainbow | TD3 |
| | n step | | 3 | 5 |
| | skill horizon | | 30 | 100 |
| | exploration noise | | 0.4 | 0.2 |
| | MLP size | | [512, 512] | |
| | optimizer | | Adam | |
| | learning rate | | $3 \times 10^{-4}$ | |
| | batch size | | 64 | |
| Graph Selection Policy | algorithm | | PPO | |
| | optimizer | | Adam | |
| | learning rate | | $1 \times 10^{-4}$ | |
| | batch size | | 1024 | |
| | clip ratio | | 0.1 | |
| | MLP size | | [512, 512] | |
| | GAE $\lambda$ | | 0.95 | |
| | entropy coefficient | | 0.1 | |
| Learned Dynamics Model | optimizer | | Adam | |
| | learning rate | | $3 \times 10^{-4}$ | |
| | batch size | | 128 | |
| | number of attention layers | | 1 | |
| | attention embedding size | | 128 | |
| | number of heads | | 4 | |
| Task Skill Selection Policy | algorithm | | PPO | |
| | MLP size | | [512, 512] | |
| | optimizer | | Adam | |
| | learning rate | | $1 \times 10^{-4}$ | |
| | batch size | | 1024 | |
| | clip ratio | | 0.1 | |
| | GAE $\lambda$ | | 0.95 | |
| | entropy coefficient | | 0.02 | |
| Training | # of random seeds | | 5 | |
| | diversity reward coefficient $\beta$ | | 0.5 | |

- bucket, soap → bucket: the water in the bucket getting soap in it.
- bucket, rag → rag: the rag getting cleaned in the bucket (which requires that the rag is dirty and the water in the bucket gets soap in it).

## E.2  2D Minecraft Results

In addtion to the environments shown in the paper, we further evaluating our method in larger-scale settings in 2D Minecraft with 15 state factors following Andreas et al. [1].

The state space (15 state factors) consits of: the agent (location and direction), 10 environment entities (the positions of 3 wood, 1 grass, 1 stone, 1 gold, and 4 rocks surrounding the gold), and 4 inventory cells (i.e., the number of stick, rope, wood axe, and stone axe that the agent has).

The action space (9 discrete actions) consists of:

- 4 navigation actions: moving up, down, left, right,
- picking up the environment entity in front, which has no effect if the agent does not have the necessary tool for collecting it,
- 4 crafting actions: crafting a stick/rope/wood axe/stone axe, no effect if the agent does not have enough ingredients.

In the downstream Mine Gold task, the agent will receive a sparse reward after finishing all the following steps

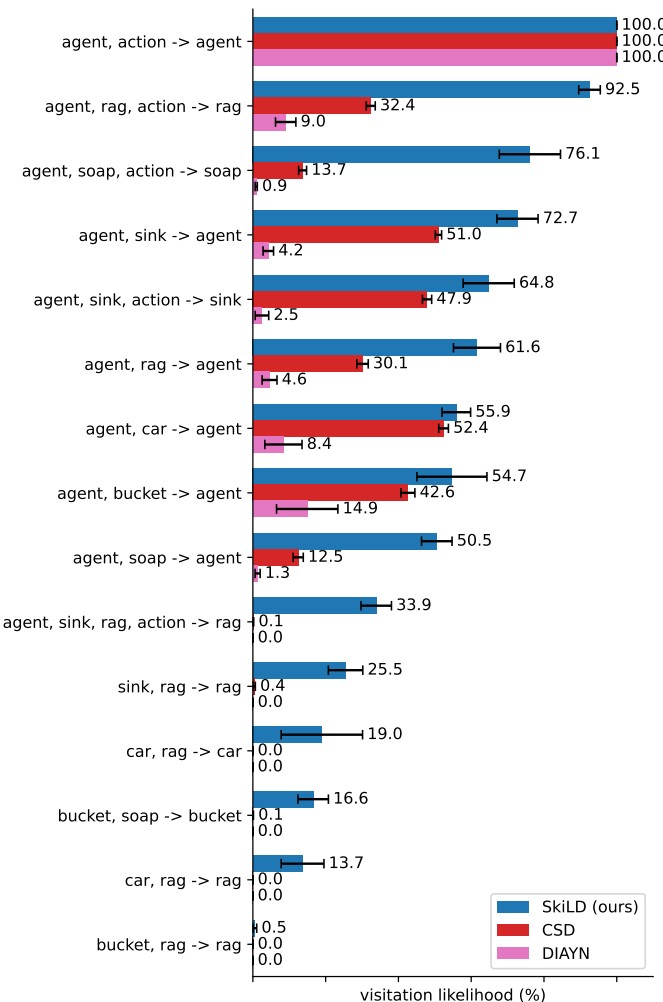

Figure 8: Among all inducible dependency graphs, the percentage of episodes where each graph is induced through random skill sampling. Standard deviation is calculated across five random seeds.

- collecting a unit of wood to craft a wood stick,
- collecting another unit of wood and combining it with the stick to craft a wood axe that is required for collecting the stone and for removing the rock,
- collecting a unit of wood and a unit of stone to craft a stick and then a stone axe that is required for collecting the gold, remove the rock surrounding the gold and collect the gold with the stone axe.

As shown in Fig. 9, SkiLD still outperforms all baselines in this complex task, demonstrating the usefulness of its learned skills for downstream task solving.

# F   Skill Visualizations

In Figure 10 we visualize three challenging long-horizon skills learned by SkiLD: thawing the olive, cleaning the car, and cutting the peach. All of these skills require a sequence of interactions that is difficult to recover without directed behavior. Thus, comparable baselines do not learn skills of similar complexity. More skill visualizations can be found at: `https://wangzizhao.github.io/SkiLD/`.

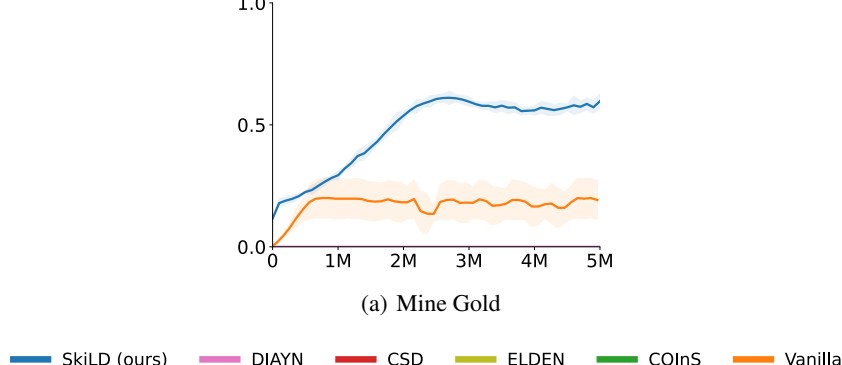

(a) Mine Gold

Figure 9: Training curves of SkiLD and baselines on the 2D Minecraft downstream task (reward supervised second phase). Each curve depicts the mean and standard deviation of the success rate over 5 random seeds. SkiLD outperforms all baselines, converging faster and to higher returns.

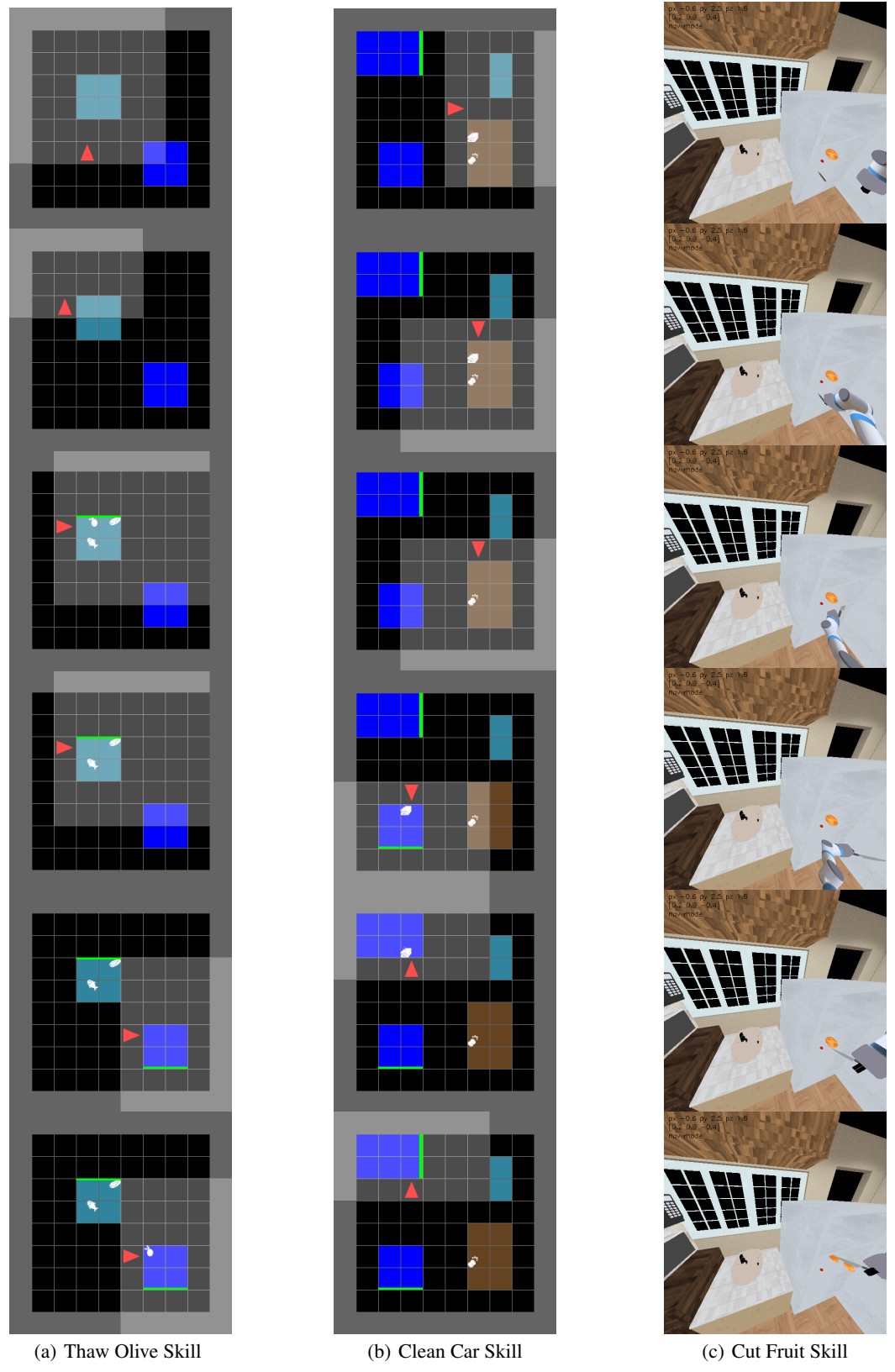

(a) Thaw Olive Skill    (b) Clean Car Skill    (c) Cut Fruit Skill

Figure 10: Policy rollouts for learned policies that achieve long horizon tasks **(a)** Mini-BH thaw olive, **(b)** Mini-BH clean car, **(b)** iGibson cut peach.

