# OpenReview forum: "SkiLD: Unsupervised Skill Discovery Guided by Factor Interactions"
_NeurIPS.cc/2024/Conference — NeurIPS 2024 poster_

### Official Review · Reviewer_WEDj · 2024-06-23

**Soundness:** 3
**Presentation:** 2
**Contribution:** 3
**Rating:** 6
**Confidence:** 3

**Summary:**

The authors present SkiLD, an unsupervised RL method that learns skills by augmenting DIAYN reward with a graph-state dependency reward to induce meaningful changes in object interactions.

**Strengths:**

**Experiments:** Experiments are performed on a reasonably comprehensive set of 10 downstream tasks in 2 domains, and demonstrate that SkiLD is significantly better than all baselines.

**Novelty and Idea:** The idea of using explicit factors as skill learning reward is interesting and novel. At a high-level, it’s also pretty intuitive.

**Weaknesses:**

**Clarity: Quite a few clarity issues**

- In the main paper, the method could be presented more clearly at a higher level. For example, introducing the Graph-Selection policy first (in 3.1 instead of 3.2) could help with understanding, at a higher level, what the method is trying to do. Currently the paper first talks about the skill selection policy but, by the time the reader gets to 3.1, it’s not fully clear yet (it’s mentioned at the top of section 3 but then gets into details a bit too suddenly in 3.1) that there is an initial unsupervised skill learning phase and what that phase is trying to do. Going from high level idea → high level policy → low level policy would make this more clear.
- More explicit details are needed. How is the dynamics model trained? What’s the high-level overview of the algorithm? When and how are diversity rewards applied? What exactly are the input/outputs features of the graph selection policy? All of this could be more clear if the authors provide some pseudocode linked to from the main paper and beef up the appendix.
- High-level details about environment assumptions (learned/given dynamics, what is given in the graph factorization, etc.) in section 4.1 should be given, even if details are already in the appendix.

**Experiments:**

- Are 10M timesteps really needed even in Mini-behavior domain? This seems extremely sample-inefficient. Same with requiring up to 5M timesteps to learn downstream tasks reasonably in these environments. Perhaps some details about why all methods need so many timesteps would be helpful.
- How are hparams selected and tuned? And for the baselines? How do we know this is a fair comparison?

**Minor issues:**

- L163: missing end of sentence period
- Fig 4: CaSk?

**Questions:**

One limitation of all unsupervised skill learning methods is that a significant portion of the learned skills are meaningless. Qualitatively, what proportion of the skills ended up being meaningful? Was it more with SKilD than baselines like DIAYN? This perhaps could be visualized by randomly sampling ~50 skills and visualizing them to see which ones are actually meaningful.

**Limitations:**

Adequately addressed.

---

> ### Author Rebuttal · Authors · 2024-08-07
>
> We would like to thank the reviewer for the valuable and constructive feedback. We are particularly excited that the reviewer finds our idea of learning skills to induce diverse interactions between state factors novel and our evaluation sound.
>
> Please find below our responses to address your concerns regarding our work:
>
> >Regarding reorganizing Sec 3
>
> We appreciate the reviewer's suggestions, and we will rearrange these sections for greater clarity.
>
> We will also add more detail to the high-level description at the beginning of the section to make it clear what the high-level and low-level inputs are:
> * the high-level policy outputs graphs and diversity parameters.
> * the low-level skills take in both and output primitive actions.
>
> >Regarding details on dynamics model training, when and how are diversity rewards applied, what exactly are the input/output features of the graph selection policy?
>
> We appreciate this comment and will add the pseudocode to the main paper (it is also attached to the **global response**). We will also add the appendix section addressing the following components,
> * Loss functions for dynamics model: line 167 mentions that the loss is the prediction error.
>
>    Specifically, we measure it as cross-entropy loss for discrete state space and mean squared error for continuous state space.
> * When and how are diversity rewards applied: as shown in Eq. 3, when the induced graph matches the desired graph, the diversity is applied as **an additional reward** to the graph reward.
> * The input/output features of the graph selection policy:
>    * As shown in Fig 2 and line 186, the input is the current state
>    * The output feature is a categorical distribution over N, where N represents the number of unique graphs in the history of seen graphs.
>
> >High-level details about environment assumptions (learned/given dynamics, what is given in the graph factorization, etc.) in section 4.1 should be given, even if details are already in the appendix.
>
> We agree with this constructive feedback and have added this information to Sec 4.1.
>
> >Why do all methods need so many timesteps?
>
> Though mini-behavior looks simple, the environments are challenging because of the following factors:
> * The state factors are **highly sequentially interdependent**, making skill learning and task learning challenging: for example, in cleaning car environments, the agent can’t clean the car until it picks up the rag, turns on the sink, and soaks the rag. These interdependencies between state factors pose great challenges to the agent’s exploration ability.
> * During the skill-learning stage, we would like the agents to learn **all possible skills** (e.g., manipulating all objects) rather than learning to manipulate a single object. Hence, it requires more samples. Furthermore, Fig. 4 shows that, with 10M timesteps, baselines still fail to learn to manipulate all objects.
> * During task learning, we use **sparse rewards**, and thus it is further challenging for agents to explore. Our results match the results in the mini-behavior paper, which shows that it takes millions of timesteps to learn the task with sparse rewards.
> * In addition, we use primitive actions, and **many actions have no effect if it’s not applicable in the current state**. So the task is especially challenging for exploration.
>
> >Regarding hyperparameters tuning.
>
> We apply grid search on the following hyperparameters for the following methods to ensure their best performance. We will add these details to the paper Appendix. Due to space limits, we leave the values we searched in the Appendix.
> * SkiLD (ours):
>    * skill-learning:
>       * low-level policy: exploration noise, skill horizon
>       * high-level policy: entropy coefficient
>    * task learning: skill selection policy’s entropy coefficient
> * DIAYN:
>    * skill-learning: low-level policy exploration noise
>    * task learning: skill selection policy’s entropy coefficient
> * CSD:
>    * skill-learning: low-level policy exploration noise, intrinsic reward coefficient, learning rate
>    * task learning: skill selection policy’s entropy coefficient, skill horizon
> * ELDEN:
>    * dynamics model: regularization lambda and its annealing, gradient cutoff threshold
>    * task policy: entropy coefficient
> * COInS:
>    * Dynamics model:
>       * Passive and active model cutoff thresholds
>       * Interaction state reweighting ratio
>    * Granger-causal score threshold
>    * Skill learning:
>       * Goal sampling and relative action sampling distance
>       * Hindsight reweighting ratio
> * Vanilla RL (PPO): entropy coefficient, learning rate, batch size
>
> >Regarding meaningfulness of learned skills.
>
> Fig 4 in the paper (full version: Fig 1 of the global response) shows the semantic meaningfulness of skills acquired by different methods (in terms of which interactions they can induce).
>
> Specifically, for each method, we randomly sample its learned skills (see details of # of skills below) and we record which interactions they induce in each episode. From the global response Fig 1, we can see that
> * SkiLD induces all inducible graphs, suggesting that it learns skills that cover all possible interactions with objects (i.e., at least 15 skills are semantically meaningful), though there may also exist redundant or less meaningful skills.
> * In contrast, for DIAYN, with a similar number of skills, most interactions it induces are agent moving (i.e., agent, action -> agent) and agent getting blocked by other objects (e.g., agent, car -> agent). It suggests that a large portion of skills are navigation skills, and they are less meaningful when solving tasks requiring inducing object interactions.
>
> the number of skills:
> * SkiLD (ours): in the Cleaning Car environment, we uniformly sample from the history of 15 seen graphs and 4 discrete diversity variables: leading to 60 discrete skills
> * DIAYN: we uniformly sample from 64 discrete skills.
>
> >Regarding typos
>
> We appreciate the detailed feedback and have corrected these typos.

---

> ### Comment · Reviewer_WEDj · 2024-08-11
>
> Thanks for the detailed response.
>
> Due to the neurips rebuttal format, I can't verify the suggested changes will be made, so I can only partially consider those in whether to increase the score. However, the plan makes sense. I can reconsider if the authors explicitly list out quotes for what will be changed.
>
> > Though mini-behavior looks simple, the environments are challenging because of the following factors:
>
> This is a reasonable description, thank you. This information however is not presented in the actual paper, with the only hint at this being: "While conceptually simple, this domain has been shown to be extremely challenging for Vanilla RL."
>
> > Fig 4 in the paper (full version: Fig 1 of the global response) shows the semantic meaningfulness of skills acquired by different methods (in terms of which interactions they can induce).
>
> This doesn't fully answer my question (though it is a helpful visualization) of "what proportion of skills end up being meaningful" as it's easier to induce these interactions in a discrete environment over a long horizon of 500 steps. Randomly sampling skills and visualizing their behavior, truly at random, in the Gibson environment (cont. control) would be a better comparison, or perhaps plotting a graph of the state distribution of the skills (again sampled at random) against x-y positions in the minigrid environments to get more information about what the policy is doing with each skill.
>
> I have currently raised my score in response to the other parts of the rebuttal.

---

> > ### Author Response · Authors · 2024-08-11
> >
> > Again, we thank the reviewer for the constructive feedback, including the new suggestions of a more detailed description of the mini-behavior environment and the visualization of state distribution! We will ensure that they are incorporated in the next version of the paper.

---

### Official Review · Reviewer_c2kz · 2024-06-28

**Soundness:** 2
**Presentation:** 3
**Contribution:** 2
**Rating:** 5
**Confidence:** 4

**Summary:**

In the presented paper the authors propose a novel skill discovery method called Skill Discovery from Local Dependencies (SkiLD). The method utilizes the concept of local dependencies to incorporate the interaction of factors in a factorized state space. A novel intrinsic reward signal is introduced to guide the skill discovery. In the skill learning phases, different skill-conditioned policies are learned for the skills and in the task learning phase these (frozen) skills are used to learn a task policy solving more complex downstream tasks. The method is evaluated in two simulation environments and compared to several baseline methods.

---
**Post-Rebuttal/Discussion**: I appreciate the discussion with the authors and their effort in clarifying open questions, my main criticism wrt presentation and evaluation are not solved though, and I remain my score of 5.

**Strengths:**

The paper tackles an interesting, challenging and important research direction: equipping agents to acquire skills autonomously and in the absence of a concrete task that can then be used in given tasks.

The main idea behind the method – utilizing local dependencies/interactions (realized with dependency graphs) and diversity – is well motivated and introduced.

Overall the paper is written well and understandable, notations are solid, and the figures are of good quality (especially Figure 2 gives a good overview of the method).

The explicit formulation of the two main research questions is appreciated and gives a good structure and focus for the evaluation.

Similar, the explanations of baselines and especially what the respective comparison highlight is a very nice. (Explanations of differences to the different related work sections as well).

Additional ablations show the effect of the main idea.

**Weaknesses:**

My main concern is related to the use of overexaggerations and mismatch in some claims and with respect to the experimental results. For example, ‘…resulting in a robust set of transferable skills’, there is no evaluation or similar regarding robustness (in fact that word only occurs in that claim). Or more prominent, the domains are introduced with ‘…a LARGE number of state factors.’, while the state factors range from only 3 to 6 (Section 4.1).

While mentioning the research questions is good, the chosen evaluations do not fit these questions very well. Q1 is about the diversity of the interactions, but Figure 4 then only shows a comparison of some ‘particular’ (how are they chosen?) interactions. For a diversity comparison a general metric comparing all found/used interactions should be compared. The number of skills is also not shown/discussed or evaluated.

Similar, Q2 is about more efficient downstream learning. Efficiency in terms of what exactly? The according evaluation in Figure 5 (and Section 4.4) measures the performance. Is performance used as a proxy for efficiency here?

As there are many details, modules and methods, it is not completely clear what is assumed to be given/known, or how certain quantities are inferred/used. For example, the events of interest (every factor in just the next_state of the s,a,s’ tuple?) or the target dependency graph (a crucial parameter, and lines 157ff mentions HER for that, but not for all cases?). What exactly does the diversity parameter do? How is the exact flow of the skill learning phase (getting the target graph, the intrinsic reward, learning the policies, and multiple policies can be learned for each skill?)? Some additional effort in clarifying such details would greatly benefit the paper.

Some critical details, like that for some setups the ground truth dependencies were used instead of the learned, are only provided and thus somehow hidden in the Appendix.

Additional comments are given in the Questions.

While computational demanding and, hence, understandable, 5 seeds/runs are quite low for comparing RL based algorithms (known problem in the community, although often neglected/ignored). Recent frameworks and metrics have been proposed tackled at these problems related to low number of runs and comparing overall algorithms performance [1]. It would be beneficial to add such metrics (e.g., performance profiles and IQM/IQR) to the paper for a better comparison of the proposed algorithm.

-----------

[1] Agarwal, Rishabh, et al. "Deep reinforcement learning at the edge of the statistical precipice." Advances in neural information processing systems 34 (2021)

**Questions:**

How can the approach deal with varying number of objects? The policies depend on the states and the state-specific graph?

Does the chosen representation allow for ‘infinite’ number of skills? How does the approach scale with increasing skill number? How does it effect the learned policies (skill and task policy)?

How are the two learning phases (skill and task) orchestrated? Is each phase done once? Or do they take turns, if so, how?

In the factored MDP, is the action space A defined over the full S, or does it take the factorization into account?

L168: ‘…we utilize a factorized lower-level policy, where there is a separate policy for each factor.’ So each skill is realized with multiple policies? How is this modelled and especially used? How are skills then chosen, does this affect the action set of the task policy?

Notation in Figure 4 (and same in text) is unclear, what does the ‘x,y -> x’ notation represent?

Figure 1 seems to be never referenced?

How are the skill policies modelled? Table 1 only give learning details, is it a neural network? What are its details? Same for other methods, the presented parameters are not complete.

What does ‘state-specific’ dependency graph mean?

Figure 6 misses the plot description (is it mean and std as well?).

**Limitations:**

Limitations are discussed, focusing on the assumption of known state factorization and accurate detection of local dependencies.

An additional (potential) limitation is related to the scaling of the approach wrt. the training (data, time, ..), the number of objects and skills (rather low currently), or the transfer of skills to novel objects.

---

> ### Author Rebuttal · Authors · 2024-08-07
>
> We would like to thank the reviewer for the valuable and constructive feedback. We are particularly excited that the reviewer finds our method well-motivated and our presentation clear.
>
> Please find below our responses to address your concerns regarding our work:
>
> >Regarding the use of overexaggerations and mismatch in some claims and with respect to the experimental results
>
> We appreciate the reviewer’s observation and will make sure to eliminate all overexaggerations. We have made adjustments to the language of the paper to better match the empirical results. In particular, we have removed any mention of “large” number of state factors or “robustness”, as well as other possible exaggerations such as “object-rich” or “realistic”, and replace with more precise language such as object-factored and 3D real-world-based simulation.
>
> >Regarding Q1 evaluation
>
> We appreciate this insightful comment and will incorporate the following clarifications into the paper.
>
> Due to space limitations, we deliberately only show task-relevant interactions in Figure 4, since these interactions are more informative. That being said, we agree with the reviewer that a general metric comparing all found interactions against all inducible interactions will better showcase the diversity of interactions for different methods. To this end, **we additionally show *all inducible interactions* and whether each interaction is induced, in the global response Fig 1**.
>
> **The diversity metric would be (# induced interactions) / (# induciable interactions)**. As shown in global response Fig 1, SkiLD induces all inducible interactions, while CSD only induces 80% and DIAYN only induces 60% of all inducible interactions.
>
> Regarding the number of skills, we will add the following clarification:
> * SkiLD (ours): in the Cleaning Car environment, we uniformly sample from the history of 15 seen graphs and 4 discrete diversity variables: leading to 60 discrete skills
> * DIAYN: we uniformly sample from 64 discrete skills
> * CSD: CSD uses continuous skill space, so there are an infinite number of skills.
>
> >Regarding efficiency in Q2
>
> We measure sample efficiency in terms of performance after a certain number of environment transitions.
> We appreciate this comment and will add this description to the paper.
>
> >Regarding details, modules, and methods
>
> We appreciate this comment and will add the following details to the manuscript. We also attach the pseudocode in the global response Alg. 1 to improve the clarity of the workflow.
>
> * Regarding the events of interest, it is known from the collected transitions.
> * Regarding the target dependency graph, it is sampled from the high-level policy and known.
> * Regarding HER, following the original paper, we apply it to 87.5% of sampled trajectories, and we will add this hyperparameter to the paper.
> * Regarding the diversity variable,
>    * Fig 1 mentions that the skill has two goals: (1) inducing the target dependency graph, and (2) inducing the graph in diverse ways (i.e., reaching different states).
>    * Lines 173 - 175 mention that the diversity variable fulfills the second goal, in the same way as DIAYN – the agent will be rewarded to visit states if the diversity variable can be distinguished from the states. Hence, the agent is trained to visit different distinguishable states under different diversity variables.
> * Regarding the flow of the skill learning phase: we describe it in the pseudocode in the global response and will include it in the paper.
>
> >Regarding other metrics
>
> We appreciate this insightful comment and have added IQM scores to the global response.
>
> >Regarding varying numbers of objects
>
> Please note that for the scope of this paper, we only consider MDPs, which are fully observable by definition, with a known state space (and therefore a known number of objects)).
>
> To handle varying numbers of objects, one possible extension is to pre-assign a large number of state factors and use placeholders for objects that are not present. The assumption is that objects that are not present do not involve interactions, and their states do not change.
>
> >Regarding allowing for ‘infinite’ number of skills and scaling to it
>
> A simple way to extend to an ‘infinite’ number of skills is to have the diversity variable be continuous instead of discrete. As the number of skills increases, the capacity of the agent’s behavior grows proportionally, allowing for more diverse skills.
>
> In the meantime, it will also be harder for the skill policy and the discriminator to process these skills, potentially hampering skill learning. Finding the “right” number of skills is often a domain-specific problem
>
> >Regarding two learning phases
>
> Each phase is done once.
>
> >Regarding the action space
>
> It is defined over the full state space.
>
> > Regarding the factorized lower-level policy
>
> Each skill is realized by one policy network.
> * As described in Appendix Sec A, there are N (N: # of state factors) parameterized low-level networks, one per state factor.
> * When the target dependency graph is specified, we will identify which state factor should be influenced and use its corresponding policy to sample primitive actions
>    * As a result, only one of the N networks is activated at every time step.
> * The task policy still selects the target graph from the history of seen graphs and thus is unaffected by this specific design of low-level policy.
>
> >Regarding Fig 4 notations,
>
> $x,y \\rightarrow x$ represents that $x_{t+1}$ locally depends on $x_t$ and $y_t$.
>
> >Regarding ‘state-specific’ dependency graph
>
> A general dependency does not necessarily happen in every state. For example, though a knife can cut fruit, when they are far away, the cutting will not happen. The state-specific dependency graph encodes this state-specific dependency between factors, the edge represents the dependency happening in a given state.
>
> >Regarding Fig 6
>
> It is also mean and standard deviation, and we will add that clarification to the final paper.

---

> > ### Comment · Reviewer_c2kz · 2024-08-09
> >
> > Thanks for detailed response that clarified most points and the provided pseudo-algorithm helps to understand the flow better. However, it will be a challenge to incorporate all this necessary information in an updated paper.
> >
> > One follow up question:
> >
> > *"The diversity metric would be (# induced interactions) / (# induciable interactions). As shown in global response Fig 1, SkiLD induces all inducible interactions, while CSD only induces 80% and DIAYN only induces 60% of all inducible interactions."*
> >
> > I do not understand how the mentioned 100% and 80% and the global response Fig 1 are connected? Fig 1 shows in how many episodes each graph was induced, except the first one which all methods finds, no other has 100%. Moroever, one graph was only induced in 0.5% according to the Fig 1. How does this fit together with the mentioned numbers? (and with 5 random seeds, how can the Figure numbers occur, shouldn't it be in steps of 20% then? the connection between seeds and episodes is unclear here)

---

> > > ### Author Response · Authors · 2024-08-10
> > >
> > > We appreciate your quick response! Please find below our responses to address your concerns regarding our work:
> > >
> > > >Regarding incorporating necessary information
> > >
> > > We thank the reviewer for the heads-up. The main space-taking thing we are planning to incorporate into the main text is the pseudocode. Since there will be one more page for the camera-ready version were this paper to be accepted. We believe we will have enough space to incorporate all feedback into the next version of our paper.
> > >
> > > >Regarding diversity metric
> > >
> > > Fig 1 in rebuttal shows all the induced interactions AND the percentage of times they appear at least once within an episode. Obviously, some interactions will be induced all the time while some hard interactions can rarely be induced, which is what the numbers reflect. The error bar shows the standard deviation of this percentage across random seeds (the number shows the mean).
> > >
> > > Now, the new metric that we are reporting in the rebuttal (following the suggestions of the reviewer) is WHETHER an interaction will appear at all through randomly sampled skills, which can be obtained by counting the non-zero entries in the chart – 15 for SkiLD, 12 for CSD, and 9 for DIAYN, which is why our method has a skill coverage of 100%.
> > > Why an interaction should count even if it is induced rarely – since we **uniformly sample skills**, it is very unlikely to fulfill the preconditions of a hard interaction and induce it. However, during task learning, one can **optimally select the sequence of skills** to induce it using planning/a learned task policy. Therefore, even if an interaction is induced infrequently in Fig 1, it shows the skills are useful for solving tasks relevant to this interaction.

---

### Official Review · Reviewer_jfpH · 2024-07-04

**Soundness:** 4
**Presentation:** 4
**Contribution:** 3
**Rating:** 8
**Confidence:** 4

**Summary:**

The paper introduces SkiLD (Skill Discovery from Local Dependencies), an unsupervised skill discovery method. Unlike existing methods that focus on state diversity, SkiLD leverages state factorization to guide skill learning by inducing diverse interactions between state factors.

This method is designed to be more effective in complex environments, such as household settings with numerous objects. SkiLD uses local dependencies to model interactions and introduces a novel intrinsic reward mechanism.

The method is evaluated in several domains, including a realistic household robot simulation, demonstrating superior performance compared to existing methods.

**Strengths:**

- The clarity, presentation and writing of the paper are great.
- The problem of unsupervised skill discovery is an important one.
- The paper presents strong empirical results, demonstrating that SkiLD outperforms other unsupervised reinforcement learning methods in various challenging tasks.
- The experimental setup is properly designed (i.e., right choice of baselines and domains).

**Weaknesses:**

Minor weaknesses:
- The method's reliance on accurately detecting and modeling local dependencies adds a layer of complexity that may limit its applicability.
- The effectiveness of SkiLD hinges on the availability of a factored state space, which may not always be available or easily obtainable.

**Questions:**

The two points raised within the weaknesses limit the applicability and thus the impact of the paper. Do you have insights on this? Especially, how the method can be made more general?

Nitpicking:
> L88: Formally, for an event of interest $Y$ and its potential causes $X = (X^1,  ..., X^N )$, given the value of $X = x$, local dependencies focus on which $X$ is are the state-specific cause of the outcome event $Y = y$.

I think the vocabulary used in this sentence is wrong.
- $Y$ is not an event but a random variable.
- $Y = y$ is an event.

> L108: In this work, for a transition $(\mathcal{S} = s, \mathcal{A} = a, \mathcal{S}^\prime = s^\prime)$...

If I understand correctly the notations, this shouldn't be $\mathcal{S}$ and $\mathcal{A}$ but $S$ and $A$, because $\mathcal{S}$ and $\mathcal{A}$ are the state space and action space and are not random variables. Plus, you said in the Background section that "In this paper, we use uppercase letters to denote random variables and lowercase for their specific values...".

Same remark for L149-150.

> In this section, we describe SkiLD, which enhances the expressivity of skills using local dependencies. SkiLD represents local dependencies as state-specific dependency graphs, defined in Sec. 2.2.

State-specific dependency graphs are actually not defined in Section 2.2, there is just a sentence at the end of the section. It would be useful to better explain what are these graphs.

**Limitations:**

Yes, the authors adequately addressed the limitations.

---

> ### Author Rebuttal · Authors · 2024-08-07
>
> We would like to thank the reviewer for the valuable and constructive feedback. We are particularly excited that the reviewer finds our presentation clear and our evaluation sound.
>
> Please find below our responses to address your concerns regarding our work:
>
> >The method's reliance on accurately detecting and modeling local dependencies adds a layer of complexity that may limit its applicability.
>
> With respect to detecting and modeling local dependencies, two possible avenues offer promising directions:
> * Relaxing the dependence of precise local-dependency identification by using a window of states rather than a per-state reward when incorporating dependency information.
> * Since many state-specific dependencies fall into broad categories (contact, co-occurrence, etc.), employing a meta-learning strategy for identifying local dependencies in a lifelong setting can improve the effectiveness of dependency identification.
>
> >The effectiveness of SkiLD hinges on the availability of a factored state space, which may not always be available or easily obtainable.
>
> With respect to factored states, we agree that SkiLD requires a factored state space. In the meantime, we believe that advances in vision, especially related work such as SAM [1] and other object-centric modeling [2] offer a path forward for extracting a factored state space from a dense state such as pixels, thereby extending the application scope of SkiLD.
>
> [1] Ravi, Nikhila, et al. "SAM 2: Segment Anything in Images and Videos." arXiv preprint arXiv:2408.00714 (2024).
>
> [2] Aydemir, Görkay, Weidi Xie, and Fatma Guney. "Self-supervised object-centric learning for videos." Advances in Neural Information Processing Systems 36 (2024).
>
> >Regarding the usage of symbols
>
> We appreciate the detailed feedback and will make sure we use the correct symbol in the paper!
>
> >Regarding the definition of state-specific dependency graphs
>
> We appreciate this insightful comment and will add definitions of state-specific dependency graphs to section 2.2 to help it better integrate into section 3.

---

> > ### Comment · Reviewer_jfpH · 2024-08-08
> >
> > Thank you for your detailed rebuttal. I will maintain my score.

---

### Official Review · Reviewer_dKfG · 2024-07-13

**Soundness:** 3
**Presentation:** 3
**Contribution:** 3
**Rating:** 5
**Confidence:** 4

**Summary:**

The paper introduces SkiLD, a novel method leveraging state factorization to guide skill learning in unsupervised reinforcement learning. SkiLD emphasizes learning skills that induce diverse interactions between state factors, which are crucial for solving downstream tasks. The authors demonstrate that SkiLD outperforms existing unsupervised RL methods through empirical validation.

**Strengths:**

1. **Novel Approach**: SkiLD introduces a unique approach to skill discovery by focusing on local dependencies and interactions between state factors, addressing the limitations of state diversity methods.
2. **Empirical Validation**: The effectiveness of SkiLD is demonstrated through experiments in various environments, showcasing its superior performance compared to baseline methods.

**Weaknesses:**

1. **Assumption of Factored State Space**: SkiLD assumes access to a factored state space, which may not always be available or easy to obtain in real-world applications.
2. **Evaluation on Limited Domains**: The evaluation domains are somewhat restricted. Broader evaluation across more commonly used environments could emphasize the method's performance. Evaluating environments like Crafter, 2D Minecraft, or manipulation environments with multiple objects would better illustrate the benefits of focusing on local dependencies.
3. **Scalability**: There is a need to explore how SkiLD scales with an increasing number of state factors, which is crucial for practical applications.

**Questions:**

Please refer to Weakness

**Limitations:**

Please refer to Weakness

---

> ### Author Rebuttal · Authors · 2024-08-07
>
> We would like to thank the reviewer for the valuable and constructive feedback. We are particularly excited that the reviewer finds our idea of learning skills to induce diverse interactions between state factors novel and our evaluation sound.
>
> Please find below our responses to address your concerns regarding our work:
>
> >Assumption of Factored State Space: SkiLD assumes access to a factored state space, which may not always be available or easy to obtain in real-world applications.
>
> With the recent advances in object segmentation and object-centric representation techniques, such as SAM [1] and SOLV [2], one potential way to construct the factored state space is to use these models to extract factored representation from real-world image observations.
>
> We agree that testing the effectiveness of SkiLD with learned factored representations would be an interesting next step, and will add it to the future work section of the next version of this paper.
>
> [1] Ravi, Nikhila, et al. "SAM 2: Segment Anything in Images and Videos." arXiv preprint arXiv:2408.00714 (2024).
> [2] Aydemir, Görkay, Weidi Xie, and Fatma Guney. "Self-supervised object-centric learning for videos." Advances in Neural Information Processing Systems 36 (2024).
>
> >Evaluation on Limited Domains: The evaluation domains are somewhat restricted. Broader evaluation across more commonly used environments could emphasize the method's performance. Evaluating environments like Crafter, 2D Minecraft, or manipulation environments with multiple objects would better illustrate the benefits of focusing on local dependencies.
> >Scalability: There is a need to explore how SkiLD scales with an increasing number of state factors, which is crucial for practical applications.
>
> We appreciate the reviewer’s suggestions and agree that further empirical testing in other commonly used or larger-scale settings is an important direction for future work.
>
> Meanwhile, we emphasize that, while exploring scaling further is valuable, we believe that it will not fundamentally change the scientific insights we introduced – compared to prior works in unsupervised skill discovery, by focusing on local dependencies between state factors, SkiLD learns skills that induce more diverse interactions and enable more sample-efficient downstream task learning, for the following two reasons:
>
> 1. The environments used in the paper are also challenging and baselines fail to learn diverse skills and solve downstream tasks.
>    * Specifically, the mini-behavior environments have the same **inter-dependency between state factors** as 2D Minecraft (e.g., in Cleaning Car, the agent cannot clean the car until it soaks the rag in the sink).
>    * For the iGibson environment, despite the four task-relevant factors, it also contains many intractable objects such as jars, microwaves, garbage cans, etc.
>
>    In contrast, **most prior works in unsupervised discovery are only evaluated in environments with one or two state factors (where skills are limited to moving the agent to different locations)**. As a result, as shown in Fig 4 and 5, when evaluated on Mini-behavior and iGibson,  baselines fail to learn diverse skills and solve downstream tasks. Compared to them, SkiLD learns to induce those complex inter-dependency graphs and solves tasks successfully.
> 2. In principle, nothing in our method prevents it from scaling to more state factors.
>    * For methods that focus on reaching diverse states (like DIAYN), due to the **exponentially growing state space**, it would be more challenging to learn to induce meaningful interactions.
>    * In contrast, **the number of inducible interactions typically increases much more slowly**. As a result, by focusing on inducing diverse interactions, our method in principle has larger advantages than existing methods.

---

> > ### Comment · Reviewer_dKfG · 2024-08-10
> >
> > Thank you for your rebuttal.
> >
> > While I appreciate the clarifications and explanations provided, I believe the current version of your paper still requires additional experimental validation to convincingly demonstrate the effectiveness of the proposed framework. I will maintain my score.

---

> > > ### Author Response · Authors · 2024-08-14
> > > **scalability experiments on 2d minecraft**
> > >
> > > We agree with the reviewer that further evaluating our method in larger-scale settings (with more state factors) is important. To do so, we evaluate our method in **2D Minecraft with 15 state factors** following Andreas et al [1], and we hope this result addresses the reviewer's concern about the evaluation and scalability of our method.
> > >
> > > Specifically, we use the **Mine Gold** task described below, measure the task success rate after **3M** time steps, and show the **IQM score** (the higher the better) as suggested by Reviewer3 c2kz across **5 random seeds**. Again, our method SkiLD outperforms the following baselines during task learning.
> > >
> > > | **Task**         | **SkiLD**         | **Elden**         | **DIAYN**         | **Vanilla**       |
> > > |-------------------|-------------------|-------------------|-------------------|-------------------|
> > > | Mine Gold     | **0.613$\pm$ 0.065**  | 0.000 $\pm$ 0.000  | 0.000 $\pm$ 0.000  | 0.000 $\pm$ 0.000
> > >
> > > [1] Andreas, Jacob, Dan Klein, and Sergey Levine. "Modular multitask reinforcement learning with policy sketches." International conference on machine learning. PMLR, 2017.
> > >
> > > We will attach the figure of task training curves in the next version of the paper (unfortunately, we can't upload figures during the discussion period). Also, due to the limit of time and computation, we will include the results of COInS and CSD in the next version of the paper.
> > >
> > > In case you are interested, the **environment details** are listed below. We make sure they will be added to the appendix.
> > >
> > > * state space (15 state factors): the agent (location and direction), 10 environment entities (the positions of 3 wood, 1 grass, 1 stone, 1 gold, and 4 rocks surrounding the gold), and 4 inventory cells (i.e., the number of stick, rope, wood axe, and stone axe that the agent has).
> > > * action space (9 discrete actions):
> > >    * 4 navigation: going up, down, left, right
> > >    * pick up the environment entity in front, no effect if the agent does not have the necessary tool for collecting it
> > >    * 4 crafting: craft a stick/rope/wood axe/stone axe, no effect if the agent does not have enough ingredients
> > > * Mine Gold task: the agent will receive a **sparse reward** after finishing **all** the following steps
> > >    * collecting a unit of wood to craft a wood stick
> > >    * collecting another unit of wood and combining it with the stick to craft a wood axe that
> > > is required for collecting the stone and for removing the rock,
> > >    * collecting a unit of wood and a unit of stone to craft a stick and then a stone axe that is
> > > required for collecting the gold,
> > >    * remove the rock surrounding the gold and collect the gold with the stone axe.

---

> > > > ### Comment · Reviewer_dKfG · 2024-08-14
> > > >
> > > > I deeply appreciate the effort put into conducting the additional experiments. Regarding the 2D Minecraft task, my concerns about limited domains and scalability issues have been resolved. I would appreciate it if the authors could incorporate these experimental results into the final version.
> > > >
> > > > Given these improvements, I will increase my score to 5.

---

### Author Rebuttal · Authors · 2024-08-07

We thank the reviewers for their thoughtful feedback and suggestions. We are particularly excited that the reviewers find our idea of learning skills to induce diverse interactions between state factors well motivated (R3), novel (R1, R2, and R4), effective (R1, R2), and sound (R1, R2, R4). We also appreciate the reviewers for commending the clarity of the technical details (R2 and R3). In this thread, we summarize our responses to the common concerns shared by the reviews. Please refer to the rebuttal attached to each reviewer's comments for a reviewer-specific response.

1. To improve the clarity of the paper, in Alg 1 of the attached pdf, we show a pseudocode of how SkiLD learns diverse skills during the skill-learning stage, and we will add it to the paper.
2. To further evaluate the diversity of each method’s learned skills, as suggested by Reviewer3 c2kz, we show **ALL inducible interactions** and whether each interaction is induced by different methods, in Fig 1 of the attached pdf. Again, SkiLD (ours) induces all inducible dependency graphs, while baselines fail to induce hard graphs with challenging pre-conditions.
3. For a better comparison between different methods on their performance during task learning, as suggested by Reviewer3 c2kz, we compute and report the IQM scores as follows:

| **Task**          | **SkiLD**         | **Elden**         | **COInS**         | **DIAYN**         | **CSD**           | **Vanilla**       |
|-------------------|-------------------|-------------------|-------------------|-------------------|-------------------|-------------------|
| Install Printer   | $0.996\pm 0.005$  | $0.981\pm 0.035$  | $0.000\pm 0.000$  | $0.000\pm 0.000$  | $0.000\pm 0.000$  | $0.886\pm 0.000$  |

| **Task**          | **SkiLD**         | **Elden**         | **COInS**         | **DIAYN**         | **CSD**           | **Vanilla**       |
|-------------------|-------------------|-------------------|-------------------|-------------------|-------------------|-------------------|
| Clean Rag         | **0.016** $ \pm $ 0.031 | 0.000 $ \pm $ 0.000 | 0.000 $ \pm $ 0.000 | 0.000 $ \pm $ 0.000 | 0.000 $ \pm $ 0.000 | 0.000 $ \pm $ 0.000 |
| Clean Car         | **0.177** $ \pm $ 0.286 | 0.000 $ \pm $ 0.000 | 0.000 $ \pm $ 0.000 | 0.000 $ \pm $ 0.000 | 0.177 $ \pm $ 0.286 | 0.000 $ \pm $ 0.000 |
| Soak Rag          | **0.960** $ \pm $ 0.031 | 0.098 $ \pm $ 0.196 | 0.000 $ \pm $ 0.000 | 0.000 $ \pm $ 0.000 | 0.000 $ \pm $ 0.031 | 0.000 $ \pm $ 0.000 |

| **Task**          | **SkiLD**         | **Elden**         | **COInS**         | **DIAYN**         | **CSD**           | **Vanilla**       |
|-------------------|-------------------|-------------------|-------------------|-------------------|-------------------|-------------------|
| Thaw Olive        | **0.223** $ \pm $ 0.157 | 0.000 $ \pm $ 0.000 | 0.000 $ \pm $ 0.000 | 0.000 $ \pm $ 0.000 | 0.000 $ \pm $ 0.000 | 0.000 $ \pm $ 0.000 |
| Thaw Date         | **0.646** $ \pm $ 0.432 | 0.000 $ \pm $ 0.000 | 0.000 $ \pm $ 0.000 | 0.000 $ \pm $ 0.000 | 0.000 $ \pm $ 0.000 | 0.254 $ \pm $ 0.000 |
| Thaw Fish         | **0.486** $ \pm $ 0.819 | 0.000 $ \pm $ 0.000 | 0.000 $ \pm $ 0.000 | 0.000 $ \pm $ 0.000 | 0.000 $ \pm $ 0.000 | 0.000 $ \pm $ 0.000 |
| Thaw any two      | **0.101** $ \pm $ 0.093 | 0.000 $ \pm $ 0.000 | 0.000 $ \pm $ 0.000 | 0.000 $ \pm $ 0.000 | 0.000 $ \pm $ 0.000 | 0.000 $ \pm $ 0.000 |

| **Task**          | **SkiLD**         | **Elden**         | **COInS**         | **DIAYN**         | **CSD**           | **Vanilla**       |
|-------------------|-------------------|-------------------|-------------------|-------------------|-------------------|-------------------|
| Peach             | **0.999** $ \pm $ 0.002 | 0.153 $ \pm $ 0.307 | 0.097 $ \pm $ 0.153 | 0.402 $ \pm $ 0.176 | 0.000 $ \pm $ 0.000 | 0.000 $ \pm $ 0.000 |
| Wash Peach        | **0.990** $ \pm $ 0.013 | 0.000 $ \pm $ 0.000 | 0.005 $ \pm $ 0.010 | 0.001 $ \pm $ 0.002 | 0.000 $ \pm $ 0.000 | 0.000 $ \pm $ 0.000 |
| Cut Peach         | **0.119** $ \pm $ 0.051 | 0.000 $ \pm $ 0.000 | 0.000 $ \pm $ 0.000 | 0.000 $ \pm $ 0.000 | 0.000 $ \pm $ 0.000 | 0.000 $ \pm $ 0.000 |

---

### Decision · Program_Chairs · 2024-09-25

**Decision:**

Accept (poster)

**Comment:**

The paper introduces SkiLD, a method for unsupervised skill discovery that uses state factorization to learn skills inducing diverse interactions between state factors. Although the reviewers found the paper interesting with good results, some of them expressed concerns about clarity in the writing and in several technical aspects related to the method (refer to comments by reviewers c2kz and WEDj). Although some of these were addressed in the authors’ responses, the reviewers expressed that the authors failed to address exactly how they would revise the paper to address all the mentioned issues. Apart from this, the reviewers pointed out inadequately justified claims pertaining to a "robust set of transferable skills" without sufficient experimental justification, and mention "large number of state factors" although only 3 to 6 state factors were used. Although the authors have promised to use more precise language, it is unclear how this would affect the overall readability of the paper. Although incomplete, the inclusion of results on the more complex 2d minecraft experiments looks promising. The reviewers also felt their concern regarding the usefulness of skills was not sufficiently addressed. Overall, the paper is relevant and interesting in the context of unsupervised skill discovery. However, it could benefit from clearer presentation, more accurate representations of the scope of the work, with the technical aspects described fully and precisely with appropriate context.